# Learning to Be Cautious

**Montaser Mohammedalamen**                     *mohmmeda@ualberta.ca*
*University of Alberta; Alberta Machine Intelligence Institute (Amii)*
**Dustin Morrill**                               *dustin.morrill@sony.com*
*SonyAI**
**Alexander Sieusahai**                          *asieusah@ualberta.ca*
*University of Alberta*
**Yash Satsangi**                          *yash.s.satsangi@gmail.com*
*Independent Researcher**
**Michael Bowling**                              *mbowling@ualberta.ca*
*University of Alberta; Alberta Machine Intelligence Institute (Amii)*

**Reviewed on OpenReview:** *https://openreview.net/forum?id=NXvGOaYExG*

## Abstract

A key challenge in the field of reinforcement learning is to develop agents that behave cautiously in novel situations. It is generally impossible to anticipate all situations that an autonomous system may face or what behavior would best avoid bad outcomes. An agent that could learn to be cautious would overcome this challenge by discovering for itself when and how to behave cautiously. In contrast, current approaches typically embed task-specific safety information or explicitly cautious behaviors into the system, which is error-prone and imposes extra burdens on practitioners. In this paper, we present both a sequence of tasks where cautious behavior becomes increasingly non-obvious, as well as an algorithm to demonstrate that it is possible for a system to *learn* to be cautious. The essential features of our algorithm are that it characterizes reward function uncertainty without task-specific safety information and uses this uncertainty to construct a robust policy. Specifically, we construct robust policies with a $k$-of-$N$ counterfactual regret minimization (CFR) subroutine given a learned reward function uncertainty represented by a neural network ensemble belief. These policies exhibit caution in each of our tasks without any task-specific safety tuning. Our code is available at `https://github.com/montaserFath/Learning-to-be-Cautious`.

## 1 Introduction

A key challenge in Artificial Intelligence (AI) is developing agents that can operate safely, avoiding harm to humans, other agents, property, or themselves. This is of particular concern in complex, dynamic environments that change over time, where exhaustive training and testing are infeasible. For example, consider a self-driving car that learns to drive through experience. The first time that the car encounters roads covered in snow, two problems arise and compound. Snow makes the environment more hazardous and gives the road a new appearance. A good response is to react by behaving cautiously, *e.g.*, driving more slowly, but current learning algorithms do not develop such an intuition. Generally, there are two common approaches for training AI agents, first, train-and-deploy where an agent is trained in an environment while receiving the reward, then deployed (evaluated) without receiving reward or adapting its policy further. Second, continual learning involves continuous interactions between the agent and the environment while always receiving the reward. In this paper, we target safety in the train-and-deploy scenario since it is the most common approach for current AI systems. Our primary concern is not whether the agent might learn quickly in novel situations during deployment or how it adapts after experiencing novel states, but rather understanding its zero-shot behavior when facing unfamiliar situations during deployment. Specifically, whether it exhibits cautious behavior in such circumstances.

---

*Work done when the author was at the University of Alberta

Existing approaches for safety in reinforcement learning (RL) often specify safe behavior via constraints that an agent must not violate (García & Fernández, 2015; Berkenkamp et al., 2017; Chow et al., 2018). Broadly, this amounts to formulating tasks as constrained Markov decision processes (CMDPs) (Altman, 1999). A CMDP can be solved using RL in a model-based (Aswani et al., 2013; Berkenkamp et al., 2017) or model-free (Achiam et al., 2017; Chow et al., 2018; Srinivasan et al., 2020) way. However, this approach requires pre-defining safe states that the agent is allowed to visit or safe actions the agent can take. Alternatively, some approaches design "safety functions" that incentivize pre-defined safe behaviors (Turchetta et al., 2016; Wachi & Sui, 2020; Turchetta et al., 2020). Approaches that require an a priori description of safety information about specific scenarios present a scaling problem, as it is generally infeasible to enumerate all potentially hazardous situations in realistic or open-ended applications. Our goals for this work are: (i) to illustrate why it would be useful to design agents that can automatically *learn* cautious behavior, (ii) to describe a series of simple tasks where success requires learned caution, and (iii) to provide an example system that learns to be cautious while showing its caution increases with training.

We summarize our contributions as follows:

1. We describe a sequence of tasks where cautious behavior is increasingly non-obvious, starting from a contextual bandit environment with an obvious cautious action, leading to environments where cautious actions depend on context and so cautious behavior must be learned, and concluding with a gridworld driving environment for which caution requires multi-step planning.

2. We show that the $k$-of-$N$ counterfactual regret minimization (CFR) (Chen & Bowling, 2012) robust optimization algorithm, using an ensemble of neural networks capturing epistemic uncertainty, can generate policies that adopt cautious behavior in all of these tasks.

3. We extend $k$-of-$N$ CFR from the finite horizon setting to continuing MDPs, proving that it is efficient and sound for continuing MDPs under reward uncertainty.

This paper is structured as follows: Section 2 presents a simple motivating example that explains the importance of learning to be cautious, followed by a discussion of related work. Section 3 introduces our algorithm. Experiments and results are shown in Section 4, and our theoretical contribution extending $k$-of-$N$ CFR to continuing MDPs is provided in Section 5. Section 6 outlines the approach's limitations and potential directions for future research. Finally, Section 7 reflects on the broader impact.

## 2 A Motivating Example

Consider a decision-making task where you are shown an image and must choose one of eleven actions. The images are hand-drawn digits from MNIST (LeCun et al., 1998), (*e.g.*, Figure 1a) and you observe a reward of +1 for selecting the action with the index matching the portrayed digit and zero otherwise, except for the eleventh action, which always yields a small reward, +0.25. Now, what do you do when the image is not of a familiar digit but is instead a novel[1] a shoe image from MNIST fashion (Xiao et al., 2017), (*e.g.*, Figure 1b) or a letter from EMNIST (Cohen et al., 2017), (*e.g.*, Figure 1c)? A natural choice

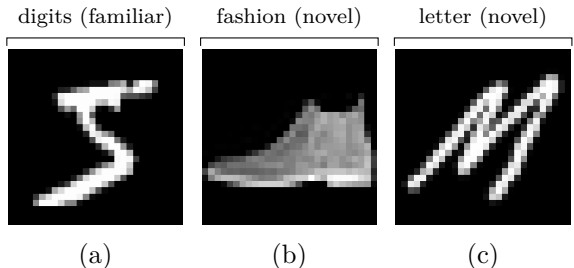

digits (familiar)   fashion (novel)   letter (novel)

(a)            (b)            (c)

Figure 1: (a) 5 from MNIST, (b) a boot from MNIST fashion, (c) "M" from EMNIST letters.

is action eleven which has always given a reward regardless of the image, while every other action often gave no reward at all. But is this choice common for current AI algorithms?

One obvious approach for choosing the next action is to guess what reward each action will yield with the new image and choose the action with the largest estimate. For example, a nearest-neighbor approach would select a previous image that resembles the new image in some way and use the rewards from the previous image as the reward estimates, effectively extrapolating from familiar to novel images. Considering that the

---

[1]"Novelty" here refers to the relative novelty due to lack of its presence in the past experiences as in Barto et al. (2013).

new image looks very different from all the previous ones, this extrapolative approach relies on a questionable premise. Algorithms like this are unlikely to choose action eleven, since its reward was always small and only one of the ten other extrapolations need to look promising for action eleven to be overlooked. A conventional RL approach like Q-learning (Watkins, 1989) or policy gradient (Williams, 1992; Sutton et al., 2000) with function approximation also employs extrapolative guessing and fails to behave cautiously in this task. In fact, after training on MNIST digits, a greedy policy with respect to a single neural network model of the reward function (effectively Q-learning) chooses action eleven less than 2% of the time when presented images from MNIST fashion (see Section 4).

A common approach to ensuring caution is to incorporate prior knowledge about safe behaviors. For example, we could designate action eleven as a "safe action" and encourage the agent to choose it when encountering a non-digit image or when it has no strong preference for any other action. Thomas et al. (2019) outlines a general methodology for such algorithms, while Kahn et al. (2017) provides a more sophisticated example. Embedding prior knowledge about safety into an algorithm would be easy and effective in this particular task, but it is not very general as safety is highly task-specific and the design burden becomes worse for complicated tasks. In this vein, we present variations on our MNIST task such that cautious behavior becomes increasingly non-obvious.

An alternative to explicitly specifying cautious behavior or safety incentives is risk-sensitive RL. Broadly, these methods characterize an agent's uncertainty about future rewards of different behaviors, and then choose *robust* behaviors, *i.e.*, those that maximize the agent's reward assuming unfavorable conditions (often with a formal risk measure). Two types of uncertainty might be present in a decision-making task, (i) *aleatoric* uncertainty that is stochasticity inherent in the environment, *e.g.*, the agent may be uncertain about the number that a die will show before it is rolled, and (ii) *epistemic* uncertainty that stems from the agent's lack of certainty about the specific environment, *e.g.*, the agent may be uncertain about a die's probability distribution, not just its outcome.

Various methods for learning policies are robust to aleatoric uncertainty (Chow et al., 2017; Tang et al., 2020; Clements et al., 2019). Still since the mapping from images and actions to rewards is deterministic in our MNIST example task, there is no aleatoric uncertainty to be robust to. Consequently, these methods do not behave differently from extrapolative systems in tasks like this. Alternatively, we could consider robustness to epistemic uncertainty. If the agent is inherently certain about action eleven's expected reward and inherently less certain about the expected rewards of the other actions, then a robust policy would choose action eleven, provided the level of uncertainty is great enough. In this case, the agent's beliefs are crafted with the domain in mind to achieve the desired behavior in much the same way as the previously discussed prior knowledge approaches. There are many more sophisticated variations on this idea (Petrik & Subramanian, 2014; Chow et al., 2015; Ghavamzadeh et al., 2016; Zahavy et al., 2021; Rigter et al., 2021). However, they share similar downsides as prior knowledge approaches. See Appendix A for more on related work, including POMDPs (Åström, 1965), distributional RL (Bellemare et al., 2017), and CMDPs (Altman, 1999).

Our approach uses robust optimization with a *learned* (i.e., posterior) belief without imposing any task-specific safety information into either component to automatically construct cautious policies. This algorithm learns autonomously to identify and choose cautious behavior that is unique to each task. We evaluate our approach in a sequence of tasks where cautious behavior is increasingly complex. This sequence begins with the previously described MNIST example and advances to a gridworld driving task that requires sequential decision-making.

## 3   Learning to Be Cautious

We have used the term caution colloquially to refer to common human behavior in unfamiliar settings, such as: (i) seeking additional oversight when available; (ii) when considering similarly promising courses of actions, preferring those with certainty about outcomes; or (iii) preferring situations where mistakes are less costly. In order to operationalize this notion into agent behavior, we propose that human notions of cautious behavior *can be well thought of as risk aversion in the face of epistemic uncertainty remaining after repeated interactions with the environment.* While notions of safety are often tied to risk aversion generally, here we

are concerned with an aversion to risk due to a lack of familiarity with novel situations (*i.e.*, uncertainty that the agent has a complete model of the world) rather than risk aversion due to stochastic outcomes.

**Markov Decision Processes.** In order to formalize the problem of learning to be cautious, we will separate the agent's "world" into the familiar and the novel, each represented as a *Markov decision process* (*MDP*). A finite, discounted MDP, $(\mathcal{S}, \mathcal{A}, p, d_\varnothing, \gamma)$, is a finite set of *states*, $\mathcal{S}$, a finite set of *actions*, $\mathcal{A}$, a Markovian *state transition probability distribution*, $p(\cdot \mid s, a) \in \Delta(\mathcal{S})$ for all states $s$ and actions $a$ (where $\Delta(\mathcal{S})$ is the probability simplex over set $\mathcal{S}$), an initial state distribution $d_\varnothing \in \Delta(\mathcal{S})$, and a discount factor, $\gamma \in [0, 1)^2$. Feedback evaluating the agent's behavior within an MDP is typically encoded as a scalar *reward function*, $r : \mathcal{S} \times \mathcal{A} \times \mathcal{S} \to [-U, U]$, where the magnitude of each reward is bounded by $U \in \mathbb{R}$. This allows us to evaluate a (stationary) policy, $\pi$ — *i.e.*, an assignment of probability distributions, $\pi(\cdot \mid s) \in \Delta(\mathcal{A})$, to each state $s$ — according to its $\gamma$-*discounted expected return*. If $S_0 \sim d_\varnothing$, $A_i \sim \pi(\cdot | S_{i-1})$, and $S_i \sim p(\cdot | S_{i-1}, A_i)$, then the normalized[3]$\gamma$-discounted expected return is $v_\varnothing(\pi; r) = (1 - \gamma)\mathbb{E}\left[\sum_{i=0}^\infty \gamma^i r(S_i, A_{i+1}, S_{i+1})\right]$. Furthermore, the algorithms we discuss will make use of $q_s(a, \pi; r) = (1 - \gamma)\mathbb{E}\left[\sum_{i=0}^\infty \gamma^i r(S_i, A_{i+1}, S_{i+1}) \mid S_0 = s, A_1 = a\right]$, which is the (normalized) $\gamma$-discounted expected return for taking action $a$ in state $s$ under reward function $r$ and discount factor $\gamma$, before following policy $\pi$ thereafter.

**Extrapolation.** Since we are primarily interested in examining the agent's *behavior* in novel situations about which they have never received feedback, *we do not define a reward function for the novel MDP*. Our primary concern is not whether the agent might learn quickly in novel situations or how it adapts after experiencing novel states and then rewards, but rather understanding its zero-shot behavior when facing the unfamiliar. Therefore, the agent does not perform any adaptation after seeing novel states, like other works (Zintgraf et al., 2020; Zhang et al., 2020; Filos et al., 2020; Rigter et al., 2021). Instead, we assess the agent's behavior in the novel MDP qualitatively, looking at the degree to which its actions in novel settings represent cautious behavior. Furthermore, we will focus on only reward uncertainty; investigating caution (and risk-aversion more generally) with transition uncertainty needs further investigation both theoretically and practically.

A straightforward approach for the agent to formulate goals for the novel MDP is to extrapolate the familiar reward function. Ordinary RL planning algorithms can then be applied to generate a policy that will perform well if the novel and familiar MDPs are very similar. Extrapolation can be done with conventional regression methods, *e.g.*, we can model the reward function as a neural network and train its parameters by applying an optimization algorithm like stochastic gradient descent to minimize the network's mean squared error. A natural approach, given such an extrapolated reward function model, $\hat{r}$, is then to behave according to an optimal policy with respect to that reward, Optim($\hat{r}$). This approach will represent a simple non-cautious baseline in our experiments.

**Inference.** A fundamental problem with extrapolation is that there are typically multiple reward models that match the familiar reward function but differ in novel situations from the novel MDP (*i.e.*, state, action, next state triples not present in the familiar MDP), even within a restricted model class. To address this issue, we can infer a posterior belief (a probability distribution) about which reward models are more reasonable, given a prior belief that describes what it means for a reward model to be "reasonable". Exact Bayesian inference is typically intractable for high-dimensional data, but a convenient approximation is to train an ensemble of neural networks, each with unique initialization parameters, and trained using stochastic gradient descent on independently shuffled familiar examples. Each neural network acts like a sample from a posterior with an implicit prior so that the entire ensemble implicitly characterizes a posterior-like belief, resulting in a larger variance in reward models output when presented with novel states compared to familiar states. Various previous works *e.g.*, Tibshirani (1996); Heskes et al. (1997); Lakshminarayanan et al. (2017); Lu & Van Roy (2017); Pearce et al. (2018); Osband et al. (2019) have used neural networks in similar ways to characterize uncertainty with connections to Bayesian inference. Alternatively, we could use methods designed to encode such a posterior compactly, such as noisy networks (Fortunato et al., 2018) or epistemic neural networks (Osband et al., 2023).

---

[2]We use the $\gamma = 0$ case to address the contextual bandit setting, such as example in Section 2.

[3]As in Kakade (2003), we use return functions that are normalized by the effective horizon, $1 - \gamma$, so that returns have the same scale as rewards. This makes optimality approximation errors easier to interpret. See Section 2.2.3 of Kakade (2003).

**Robust Optimization.** An inference approach characterizes the agent's uncertainty about what reward functions are reasonable in the novel MDP given the experience from the familiar MDP, but ordinary RL algorithms can not make use of this information beyond optimizing for a single reward function generated from the belief (*e.g.*, a sample, the expected posterior, or the maximum a posteriori reward function). However, robust policy optimization algorithms are designed to learn policies that are robust to such uncertainty. The *k-of-N CFR* algorithm computes an approximate $\mu_{k\text{-of-}N}$-robust policy, which is a policy that approximately minimizes the $k$-of-$N$ risk measure, $\mu_{k\text{-of-}N}$ (Chen & Bowling, 2012). This Bayesian risk measure is a smooth generalization of CVaR measure. By tuning the $k > 0$ and $N \geq k$ parameters, the algorithm designer can set a desired robustness level between worst-case ($N \gg k$) and average-case ($k = N$). As $N$ increases, $\mu_{k\text{-of-}N}$ approximates the CVaR measure at the $k/N$ percentile. $k$-of-$N$ CFR works by iteratively sampling $N$ reward functions from a belief and updating the current policy to improve its value under the average of the $k$-worst rewards. Importantly, the computational cost of using ensembles in this framework scales with the number of reward models $N$ and the number of $k$-of-$N$ iterations, not with the size of the state space.

As originally presented, $k$-of-$N$ CFR assumes that the uncertainty distribution is given. Another limitation of $k$-of-$N$ CFR that we overcome in this work is that it has only been described for fixed horizon MDPs, *i.e.*, those that terminate after a fixed number of decisions. We show how it can be applied in any continuing MDP, but we defer these details to Section 5 in favor of experimental results that first illustrate the utility of learning to be cautious.

**Complete Algorithm**. First, we train multiple reward models with different initializations, constructing an ensemble on the familiar MDP. Each reward model takes a state as an input and outputs the reward for each action. The ensemble of reward models produces a *learned* distribution (*e.g.*, posterior). Second, we optimize CVaR using $k$-of-$N$ CFR. At every iteration and given a state (from a familiar or novel MDP), $k$-of-$N$ CFR selects the average of the least $k$ rewards from $N$ randomly selected rewards and calculates the immediate regret given the current policy until convergence. Then it updates the policy using regret matching (Hart & Mas-Colell, 2000) with the accumulated regrets. Optimizing CVaR leads to *learned* cautious behavior. A practitioner should choose the values for $k, N$ parameters, as some applications may require a higher or lower degree of caution. A description of the complete algorithm is shown in Algorithm 1.

The experiments in the next section show that $k$-of-$N$ CFR under a neural network ensemble belief can effectively learn to be cautious in various tasks, investigating Contribution 1 and  2.

---

**Algorithm 1** Learning to Be Cautious

    **Initialize** $N$, $k$ & number of iterations $T$.
    **Train** an ensemble of $N * T$ reward models $R(r|X)$ on the familiar MDP
    **Input:** An image or a dataset of images $X$
    **Initialize**: Total Regret $P^T = 0$ & random policy $\pi_0(a|X)$
    **for** $t = 0$ **to** $T$ **do**
        Sample $N$ reward models from the ensemble & get reward predictions $R_n$
        Estimated state value: $V_\pi(X) = \sum_{i=0}^{m} R_n(r|x_i) * \pi_t(a|x_i)$
        Calculate the mean of $k$ least reward models $W_\mu = \frac{1}{k} \sum_{j=0}^{k} R_j(r|X)$
        Calculate immediate regret: $P^t = W_\mu - \sum_{a=0}^{|A|} \pi_t(a|X) * W_\mu$
        Total Regret: $P^T = P^T + \max(0, P^t)$
        Update policy: $\pi_{t+1}(a|X) = \frac{P^T}{\sum_{a=0}^{|A|} P^T}$
    **end for**

---

## 4  Experiments

We introduce a sequence of tasks designed to encourage agents to autonomously learn cautious behavior. Tasks vary in difficulty from one that requires no sequential reasoning and includes a universal cautious action, to one that requires sequential reasoning and where the return from each action is context-dependent, with a natural progression in-between. Additionally, we demonstrate that $k$-of-$N$ policies exhibit increasing

caution as the $k/N$ ratio decreases, (*i.e.*, with more risk aversion)[4]. We compare our approach to the Optim($\hat{r}$) baseline, which represents the average performance of an ensemble, in which each neural network is trained to fit the return (like traditional RL). While POMDPs (Åström, 1965), Bayesian RL (Dearden et al., 1998), or distributional RL (Bellemare et al., 2017) may be considered as alternative baselines, these are not applicable in our experiments, as our deterministic environments lack aleatoric uncertainty and are fully observable. Experimental design details and hyperparameter tuning for the algorithms are provided in Appendix B.

**Learning to Ask for Help.** Our first task is the previously described decision-making task with MNIST images in Section 2. The familiar states are the 60K training images in the MNIST digit dataset, where the initial state and each next state are sampled uniformly at random. Ten actions correspond to a digit label, and a reward of $+1$ is given when the label matches the image and zero otherwise. The eleventh action can be thought of as an "ask for help" option that always receives a reward of $+0.25$[5]. All action labels are solely to aid our discussion, whereas the agent only observes action indices. As $\gamma = 0$, the agent's return is simply the reward, making this a contextual bandit task. We represent each $k$-of-$N$ CFR instance with the last policy generated after 100 iterations.

We construct novel MDPs from the MNIST fashion test set (Xiao et al., 2017) and EMNIST letters test set (Cohen et al., 2017) (lower and uppercase). Using the images as states, we construct two novel MDPs with two different state distribution schemes representing two evaluation scenarios. The first scenario replicates the dynamics of the familiar MDP in that each image is sampled uniformly. This describes a task where the agent must come up with a policy that works well on all novel images, without emphasizing the performance on any particular one, which we call the *all-images regime*. Our second scenario uses a point-mass initial state distribution and identity transition distribution. This scenario corresponds to a decision-making task where a single crucial novel state is given instead of a distribution over multiple possible novel states, which we call the *single-image regime*. In this scenario, the impact of robustness is exaggerated because the $k$-of-$N$ CFR policy trains on the $k$-worst reward functions specifically targeted to a single state rather than the $k$-worst averaged across many states. Figure 2a shows the results of both experiments, where the individual points within each $k$-of-$N$ bar show the value of ten individual runs, while the bar height marks the average value. The solid horizontal line for Optim($\hat{r}$) represents the average across all 2K neural network reward functions in the ensemble belief, and the violin shows the variance.

In both state distribution regimes, the classification accuracy of all policies, including the most robust $k$-of-$N$ policies, on the MNIST digit test set, ranges from 96.9% to 99.4%, so even the most robust policy does not sacrifice much in accuracy. Furthermore, even the two most robust policies, 1-of-20 and 1-of-10, choose the help action less than 2% of the time when presented with familiar images. The rest of the policies, on the other hand, almost never choose the help action.

In novel settings, the "all fashion images" scenario replicates our motivating example and shows that the help action is utilized more on the fashion images by $k$-of-$N$ policies as $k/N$ is decreased (*i.e.*, with more risk aversion), up to 29% for 1-of-20. The Optim($\hat{r}$) baseline, which represents the traditional approach that ignores caution, is the least likely to use the help action on each novel dataset. We see a similar but more muted effect on the letter images, where the help action frequency ranges from 3% to 6%. Nevertheless, in the single-image regime, 1-of-20 selects the help action 89% of the time on the fashion images and 68% on the letter images — 46 and 69 times more often, respectively, than the Optim($\hat{r}$) baseline. And when 1-of-20 does not select the help action with the letter images, it does so for letters that resemble digits, *e.g.*, o, s, i, l, j, and z resemble 0, 5, 1, and 2, respectively. See Appendix B for more details, including confusion matrices of selected actions.

**Discovering Non-Obvious Cautious Actions.** In this task, an image is presented as before but there are only ten actions, and the reward for action indexed as $a \in \{0, \dots, 9\}$ is $(a + 1)$ if $a$ is the correct label for a given digit image or $-(a+2)/9$ otherwise. The reward for a correct classification increases with the action index, but so does the cost of misclassification, while keeping the mean constant under a uniform digit distribution. As a result, policies that choose lower index actions can be thought to be more cautious, avoiding the possibly large negative rewards of misclassification. Hand-crafted methods explicitly designed

---

[4]Our code is available at this GitHub repository: `https://github.com/montaserFath/Learning-to-be-Cautious`.
[5]We also conducted a sensitivity study of the eleventh action's reward ranges in $(0.1, 0.25, 0.5)$ as shown in B.1.3

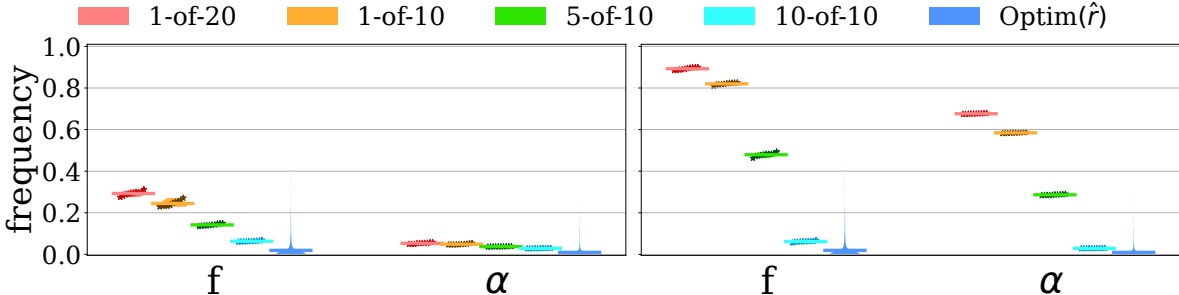

(a) Average frequency of the help action in (left) the all-images and (right) the single-image regimes.

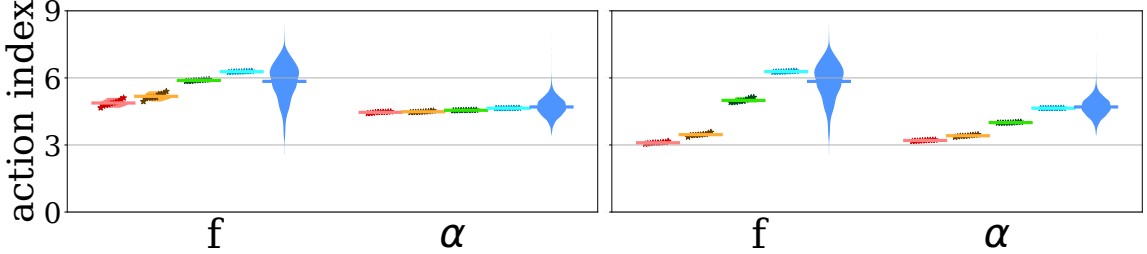

(b) Average action index chosen in (left) the all-images and (right) the single-image regimes.

Figure 2: Results for the "learning to ask for help" and "discovering non-obvious cautious actions" tasks in novel environments ("f" for fashion and "$\alpha$" for letters). Each star represents a single run, and the violin represents the variance across 10 runs.

to detect novel or out-of-distribution states, like (Vyas et al., 2018), cannot be employed in this and the following tasks because there is no designated cautious action. Again, we evaluate our approach in two regimes, one where the set of novel states is a test set and another where evaluation is done on each of these images individually. Figure 2b shows the average action index chosen by each algorithm in the novel environments.

Again, while not shown, all policies correctly label nearly all familiar test digits from MNIST images. In both regimes, on fashion images, we see that 1-of-20 and 1-of-10 systematically choose smaller indices on average than non-robust algorithms, and this increases with the $k/n$ degree of robustness. The differences are smaller on the letter images in the all-images regime, likely due to many similarities between letter and digit images, but the ordering of methods according to robustness is preserved in both regimes.

**Ask for Help Only When it is Available.** In previous tasks, cautious actions could be identified without considering input features. However, in this task, we modify the "discovering non-obvious cautious actions" task by adding the "help" action, whose value depends on an additional input feature. This feature is a binary signal that signals the availability of help. When the signal is true, the "help" action receives a reward of $+1/20$[6], otherwise it receives a reward of $-11/9$. The "help" action is better than any incorrect classification and worse than correctly classifying even the least valuable digit (zero) if help is available. If help is unavailable, the "help" action is the worst action as it always receives $-11/9$. Figure 3a shows the results for each policy in the all-images (left) and single-image regimes (right). With both fashion and letter images, we see that the robust methods with $k < N$ select the help action much more than the non-robust methods when help is available. When help is unavailable, these methods never select the help action and instead choose actions with smaller indices. The average action index decreases much more when help is available because policies switch from choosing actions with high indices to choosing the help action.

**How Caution Depends on the Extent of Training Data?** Do $k$-of-$N$ policies really *learn* to be cautious? Here, we investigate how our cautious algorithms behave with more or less training data. We repeat

---

[6]We also conducted a sensitivity study of the "help" action's reward ranges in $(+1/50, +1/20, +1/5)$ as shown in B.1.3

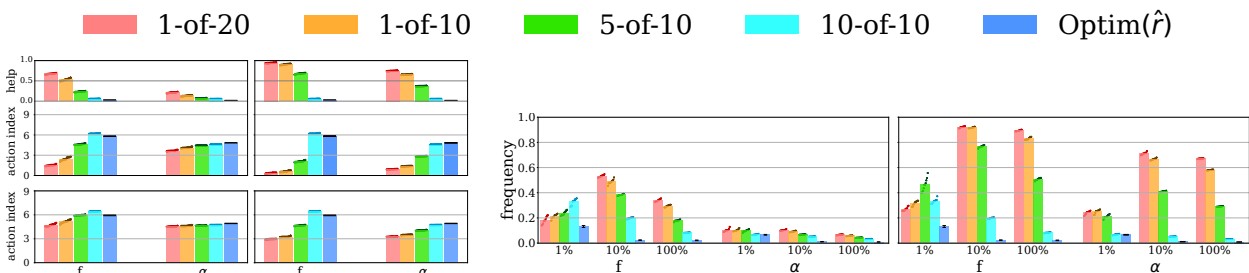

(a) Average action index and help action frequency chosen by each method in each novel environment in the "ask for help only when it is available"

(b) Average frequency of the help action in each novel environment on the "learning to ask for help" task with perturbed rewards where reward models are trained on $[1\%, 10\%, 100\%]$ of the digit dataset

Figure 3: (Left) Results for the "ask for help only when it is available" task. (Right) Results for the "learning to ask for help" task. "f" for fashion and "$\alpha$" for letters. Each star represents a single run.

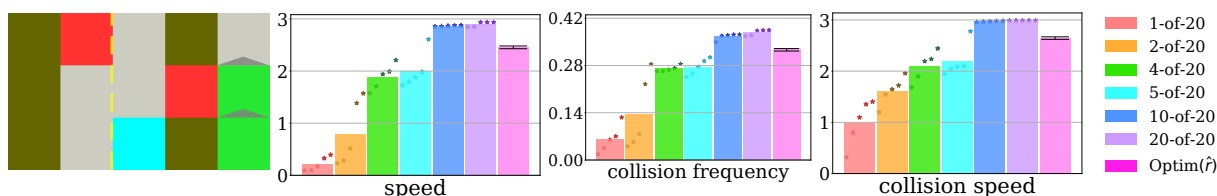

Figure 4: Left: a frame from the driving gridworld environment. The car is the cyan square, the obstacles are red, and the speedometer is green. Right: normalized $\gamma$-discounted safety statistics for each algorithm in the driving gridworld, where each star represents a single run.

the "learning to ask for help" task except that rewards are perturbed by white noise (standard deviation 0.1) before training, and the training data are $[1\%, 10\%, 100\%]$ of the full digit dataset[7]. In Figure 3b, when reward models are trained with $1\%$ of the digit images, we observe that decreasing $k$ to increase robustness does not induce caution. Effectively, the neural network ensemble belief has not seen enough data to infer that the "ask for help" action yields a small but consistent reward. Increasing the training set size to $10\%$, the correlation between robustness and caution returns is even stronger than when training with the full dataset. This shows that caution requires enough training data for the agent to accurately infer the training reward function, and once achieved, the robust agents can identify cautious behavior.

**Driving Gridworld.** For a more complex sequential decision-making task, to thoroughly evaluate Contribution 1 and 2, we introduce a gridworld driving environment (see Figure 4), inspired by AI safety gridworlds (Leike et al., 2017). The state is a five-column image: two center columns for a two-lane road, two outer columns for road shoulders, and the last column as a speedometer displaying values from 0 to 3. The agent's car is at the bottom, with the world shifting downward as it moves forward. The image height determines the agent's vision range. To limit state complexity, only one obstacle can exist on each half of the gridworld at a time, and the vision range is two. The speed limit is set to the vision range plus one, allowing the agent to "overdrive" its vision by one unit. The agent has five actions: change lane left, change lane right, accelerate, brake, and cruise. Acceleration and braking adjust speed by one unit, affecting future movement. Lane changes require momentum. Cruising maintains the current lane and speed. The goal is to maximize distance traveled, as the task is a continuing MDP. Rewards are $+1$ per forward movement, $-2$ per off-road space, and $-2$ times the car's speed for hitting an obstacle.

We investigate how our algorithm reacts to novel situations by restricting obstacles to off-road columns in the familiar MDP and allowing them to appear on the road in the novel MDP. Figure 4 shows that more robust policies drive slower, and drive over obstacles both less frequently and at slower speeds in the novel

---

[7]Noise ensures that multiple training samples are needed to learn that the help action's expected reward is constant.

MDP, reflecting intuitively cautious driving behavior. Meanwhile, the non-robust policies almost always drive at full speed, matching their behavior with the familiar MDP.

Why do we see this difference? Since obstacles are never observed on the road in the familiar MDP, there is no clear signal that driving over these obstacles will cause a bad outcome. There is a clear signal, however, that driving fast on the road yields larger rewards, so the non-robust policies optimize their behavior around this signal. The robust policies instead take the belief's uncertainty about what could happen when the car drives over an obstacle on the road into account. Since there are some reward functions in the ensemble belief that generalize collisions off-road with collisions on the road, the agent learns to avoid collisions in the novel MDP, and prefers slower speeds as there are lower worst-case consequences.

Our experiments showed a set of simple tasks where it is easy to observe cautious behavior, but still, traditional methods fail to exhibit such behavior. We then showed a proof-of-concept algorithm based on ensembles and robust policy optimization to demonstrate learned cautious behavior. Consequently, the findings support Contribution 2.

## 5 $k$-of-$N$ CFR for Continuing MDPs

The Driving Gridworld is a continuing MDP, yet $k$-of-$N$ CFR was first introduced only for the finite horizon setting. This section provides the necessary theory to extend $k$-of-$N$ to the continuing setting. CFR (Zinkevich et al., 2008) is an iterative policy improvement algorithm that uses *no-regret* learning. The regrets in CFR are "counterfactual" because, for each abstract *decision point* (something like an MDP state), $s$, we imagine modifying the CFR agent's policy in two ways: modifying to maximize its probability of reaching $s$ and further modifying it to play a given action at $s$. We compute the expected return of choosing each action at $s$ under the policy that plays to reach $s$ and apply the CFR agent's policy afterward (*counterfactual values*). The *counterfactual regret* of action $a$ at decision point $s$ is the immediate counterfactual-value advantage of $a$ compared to the policy that plays to reach $s$ and applies the CFR agent's policy there. CFR alternates between computing counterfactual regrets (policy evaluation) and applying a no-counterfactual-regret learning rule at each decision point (policy improvement). CFR's theoretical properties were originally proven in the extensive-form game framework, where decision points are analogous to *state histories* in a *fixed horizon* MDP, exponentially expanding the computation and storage required for CFR policies compared to state-based policies. Chen & Bowling (2012) shows that as long as there is no transition uncertainty, it is safe to instead implement CFR for fixed horizon MDPs where decision points are state–time-step pairs, and it can be used to compute an approximately optimal policy under the $k$-of-$N$ robustness measure.

We extend Chen & Bowling (2012)'s results to show that, when there is only reward uncertainty, it is actually safe and convenient to define decision points as states alone. We achieve this result by implementing $k$-of-$N$ CFR in terms of expected returns rather than counterfactual values. For the rest of this section, assume that all MDPs have reward *uncertainty* and transition *certainty*.

Let $v_s(\pi; r) = \mathbb{E}_{A \sim \pi(\cdot|s)}[q_s(A, \pi; r)]$ denotes the (normalized) $\gamma$-discounted expected return of policy $\pi$ from state $s$ and $\rho_s(a, \pi; r) = q_s(a, \pi; r) - v_s(\pi; r)$ is the (normalized) expected *advantage* of choosing action $a$ over $\pi$ in $s$. $d_s : s'; \pi \mapsto (1 - \gamma)\mathbb{E}\left[\sum_{i=0}^{\infty} \gamma^i \mathbb{1}\{S_i = s'\} \mid S_0 = s\right]$, where $A_i \sim \pi(\cdot|S_{i-1})$ and $S_i \sim p(\cdot|S_{i-1}, A_i)$ for $i \geq 1$, is the $\gamma$-discounted future state distribution induced by $\pi$ from initial state $s$. Kakade (2003)'s performance difference lemma for this setting is:

**Lemma 1.** *The full regret for using stationary policy $\pi$ instead of stationary competitor policy $\pi'$ from state $s$ in MDP $(\mathcal{S}, \mathcal{A}, p, d_\varnothing, \gamma)$ under reward function $r$ is*

$$v_s(\pi'; r) - v_s(\pi; r) = \frac{1}{1 - \gamma}\mathbb{E}[\rho_S(A, \pi; r)], \text{ where } S \sim d_s(\cdot; \pi') \text{ and } A \sim \pi'(\cdot|S).$$

From Lemma 1, we derive a new regret and optimality bound for CFR and $k$-of-$N$ CFR, respectively, in continuing MDPs. Given a sequence of reward functions, $(r^t)_{t=1}^T$, CFR produces a sequence of policies, $(\pi^t)_{t=1}^T$, ensures the cumulative advantage of each action $a$ at each state $s$ grows sublinearly, *i.e.*, $\sum_{t=1}^T \rho_s(a, \pi^t; r^t) \leq C^T \in o(T)$ for bound $C^T$ depends on the state-local learning algorithm used. For example, it may use *regret matching* (Hart & Mas-Colell, 2000) instances at each state to learn from $q_s(\cdot, \pi^t; r^t)$

and generate $\pi^t(\cdot \mid s)$ getting a bound of $C^T = 2U\sqrt{|\mathcal{A}|T}$. Combining with Lemma 1, arriving at CFR's cumulative full regret bound (proofs in Appendix C).

**Theorem 1.** *CFR bounds cumulative full regret with respect to any stationary policy $\pi$ as*

$$\sum_{t=1}^{T} v_\varnothing(\pi; r^t) - v_\varnothing(\pi^t; r^t) \leq C^T/(1-\gamma).$$

Taking into account Monte Carlo reward function sampling, $k$-of-$N$ CFR thus inherits the following regret bound:

**Theorem 2.** *With probability $1 - p$, $p > 0$, the full regret of $k$-of-N CFR with respect to any stationary policy, $\pi$, is upper bounded by*

$$\frac{C^T}{1-\gamma} + 4U\sqrt{2T \log 1/p}.$$

Finally, our theoretical inquiry culminates in the following optimality approximation bound for $k$-of-$N$ CFR policies:

**Theorem 3.** *With probability $1 - p$, $p > 0$, the best policy in the sequence of policies generated by $k$-of-N CFR, $(\pi^t)_{t=1}^T$, is an $\varepsilon^T$-approximation to a $\mu_{k\text{-of-N}}$-robust policy where*

$$\varepsilon^T = \frac{C^T}{(1-\gamma)T} + 4U\sqrt{\frac{2\log 1/p}{T}}$$

*and with probability at least $(1-p)(1-q)$, $q > 0$, a randomly sampled policy from this sequence is an $\varepsilon^T/q$-approximation to a $\mu_{k\text{-of-N}}$-robust policy.*

Thus, as long as a no-regret algorithm is deployed at each state in $k$-of-$N$ CFR so that $C^T$ grows sublinearly with $T$, the sequence of policies generated by $k$-of-$N$ CFR converges to a $\mu_{k\text{-of-N}}$-robust policy with high probability, as stated in Contribution 3. The proofs are in Appendix C.

## 6 Conclusions and Future Work

Our algorithm based on a neural network ensemble and $k$-of-$N$ CFR, shows that agents can learn to be cautious. Our testbeds are simple, they capture key aspects of AI safety, and they facilitate experimental comparisons. We hope that algorithms that learn to be cautious can improve the safety of, and our confidence in, deployed AI systems. There are several useful directions for future work. First, a critical limitation of our $k$-of-$N$ CFR implementation is that it is tabular and requires exact policy evaluation to determine the worst-$k$ reward functions. CFR has been used with function approximation (Waugh et al., 2015; Morrill, 2016; Brown et al., 2019; D'Orazio et al., 2020; Steinberger et al., 2020; D'Orazio, 2020) and approximate worst-case policy evaluation (Davis, 2015), applying these enhancements could scale our approach to more complicated environments. Second, transition certainty is a strong assumption that will need to be relaxed for additional applications. The increased difficulty of computing robust policies or even minimizing regret with transition uncertainty is discussed by Chen & Bowling (2012) and Even-Dar et al. (2005). It appears an algorithm must search through policies that condition on the entire state history to be sound, which makes policies complex in typical environments. Both theoretical and experimental work, such as weakening the notion of regret as in Morrill et al. (2021), is required to overcome this hurdle.

## 7 Broader Impact

We hope that cautious-learning algorithms will enhance the safety and reliability of deployed AI systems. However, automated safety measures are meant to complement, *not replace*, human judgment and safety planning. Ultimately, human oversight remains essential to ensuring safety. Moreover, while adversarial manipulation of uncertainty or concealment of reward-relevant features could induce excessive conservatism, our framework is designed to drive the agent to act cautiously.

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

# Supplementary Materials

## A  Related Work

Our work builds on a growing body of research in robust and safe RL, particularly methods leveraging CVaR to mitigate undesirable outcomes under uncertainty. While we share with prior work the motivation to develop policies that are robust to risk and uncertainty, our problem formulation introduces a distinct setting based on epistemic, not aleatory, uncertainty. Furthermore, in our setting, the agent *learn* to act cautiously without prior knowledge or constraints.

In **partially observable MDPs (POMDPs)**, the agent does not observe the environment state, instead the agent should learn to infer the state from its observations (Åström, 1965). In contrast, our approach assumes full state observability during both training and deployment. Since learning does not occur after deployment, rewards can be considered unobserved. Cautious behavior during deployment would then seek to avoid states and actions whose value has epistemic uncertainty, as this uncertainty cannot be resolved post-deployment. This is in contrast with POMDPs, where all uncertainty is aleatoric and can be resolved through further observation. For instance, in the classic Tiger POMDP (Cassandra et al., 1994), a tiger is randomly selected to be behind one of two doors, and cautious behavior might be to listen for the tiger behind the door before opening, as the listening observation can resolve the uncertainty. However, this paper focuses on fully-observable state settings, where uncertainty comes from facing novel situations and caution is about epistemic uncertainty.

In **distributional RL** instead of estimating the expected return, a full distribution over returns is estimated, aimed to capture stochasticity in rewards and transitions (Dearden et al., 1998). This distribution can be used in policy evaluation and control settings (Bellemare et al., 2017), but maybe most relatedly, it has been used to achieve risk-sensitive robustness by optimizing conditional value at risk (CVaR) in non-stationary environments where the reward is stochastic (Morimura et al., 2010). The distribution over the expected return can quantify both aleatoric and epistemic uncertainties; however, it does not differentiate between them, and so such risk sensitivity would avoid all stochastic outcomes, not just those that are unfamiliar. Similarly, **Risk-averse RL** characterizes an agent's uncertainty about future return and then chooses *robust* behaviors, *i.e.*, those that maximize the agent's reward assuming unfavorable conditions (often with a formal risk measure). There are various methods for learning policies that are robust to aleatoric uncertainty (Chow et al., 2017; Tang et al., 2020; Clements et al., 2019), which again applies risk sensitivity to the wrong category of uncertainty. Furthermore, since the mapping from images and actions to rewards is deterministic in our MNIST example task, there is, in fact, no aleatoric uncertainty to be robust to.

**Constrained MDPs (CMDPs)** (Altman, 1999) specifies safe behavior via constraints that an agent must not violate (García & Fernández, 2015; Berkenkamp et al., 2017; Chow et al., 2018). A CMDP can be solved using RL in a model-based (Aswani et al., 2013; Berkenkamp et al., 2017) or model-free (Achiam et al., 2017; Chow et al., 2018; Srinivasan et al., 2020) way. However, these approaches require pre-defining safe states that the agent is allowed to visit or safe actions the agent can take. Some approaches design "safety functions" that incentivize pre-defined safe behaviors (Turchetta et al., 2016; Wachi & Sui, 2020; Turchetta et al., 2020), or a prior over the reward in imitation learning (Javed et al., 2021). Similarly, (Yu et al., 2022; Liu et al., 2022) operate under CMDPs framework, assuming known constraints and observable rewards during training and evaluation. These methods aim to minimize constraint violations or optimize under constraint satisfaction. Compared to CMDPs, our work aims to make the agent learn to behave cautiously when facing novel situations without relying on explicit constraints. Approaches requiring a priori description of safety information about specific scenarios present a scaling problem, as it is generally infeasible to enumerate all potentially hazardous situations in a realistic application. CMDPs can be used as an approach for achieving a cautious policy with expert domain knowledge, but in no sense is it *learning* it.

In **Knows What It Knows (KWIK)**, given a state-action-next state tuple, the agent should predict the probability of the transition if the agent has observed it sufficiently (known) or $\perp$ if it has not observed

it sufficiently (unknown) (Li et al., 2008). Therefore, KWIK does not know what to do when facing the "unknown". KWIK can be viewed as a classifier to distinguish between familiar and novel states, which can then be used to hand-craft a cautious policy *e.g.*, "action eleven" in the MNIST example. However, as with CMDPs, KWIK does not *learn* this cautious behavior.

Several prior methods incorporate the **CVaR risk measure** into policy optimization, such as (Ying et al., 2022). However, their formulation defines risk solely over returns within a single fixed training environment, capturing only aleatoric uncertainty. Their subsequent evaluation explores whether optimizing for risk in aleatoric uncertainty helps with environment perturbations (transition or observation changes), which can be thought of as adding epistemic uncertainty. Their results largely showed little induction of caution under perturbation scenarios due to optimizing for only aleatoric uncertainty. Similarly, (Yang et al., 2021) uses CVaR to temper pessimism in worst-case optimization, but assumes that rewards remain defined and learnable during adaptation. In contrast, we focus on epistemic uncertainty stemming from a lack of knowledge in deterministic environments, where the agent encounters novel MDPs (e.g., MNIST vs. Fashion-MNIST) in which rewards are entirely undefined at test time.

Overall, while these works focus on robustness under aleatoric uncertainty or predefined constraints, our approach addresses epistemic-safe reinforcement learning by enabling agents to act cautiously in novel environments without access to reward or constraint supervision. We emphasize zero-shot generalization, where agents must navigate novel tasks with entirely undefined rewards.

## B  Experiments

In all experiments, $k$-of-$N$ CFR is implemented with regret matching (Hart & Mas-Colell, 2000), which is deterministic and hyperparameter-free. However, since $k$-of-$N$ CFR requires sampling $N$ reward functions, its output policy is random. Each of the $2,000$ trained neural network reward function models represents a single sample from an implicit belief, so sampling $N$ of them consists of pulling $N$ of these reward function models out of a queue. To account for the random variation caused by the ordering of the reward function models in the queue, we run multiple repetitions of $k$-of-$N$ CFR by shuffling the order of the reward function models in the queue before the start of each run. We run 10 repetitions in each MNIST experiment and 5 in the driving gridworld experiment. PyTorch (Paszke et al., 2019) is used to build and train all neural networks.

### B.1  MNIST Experiments.

For MNIST experiments, we tested three neural network architectures. One used four fully connected layers separated by rectified linear unit (ReLU) activations and the second used two convolutional layers each with one output channel, followed by two fully connected layers. These architectures were outperformed by one that begins with two convolutional layers and ends with three fully connected layers, all separated by ReLU activations. The first convolutional layer has a single input channel and 64 output channels with $4 \times 4$ kernel followed by $2 \times 2$ max-pooling. The second convolutional layer is the same except it has only 16 output channels. The fully connected layers have 50, 15, and 10 outputs, respectively. All results use this architecture.

Networks are trained to minimize the mean-squared error (MSE) between reward predictions and target rewards with the Adam optimizer (Kingma & Ba, 2015) using a learning rate of 0.0016 (we also try 0.01, and 0.001). The remaining parameters for Adam in PyTorch ($\beta_1$, $\beta_2$, $\epsilon$, and weight decay) are set to their defaults ($0.9$, $0.999$, $10^{-8}$, $0$) without the AMSGrad (Reddi et al., 2018) modification. In the "discovering non-obvious cautious actions" and "ask for help only when it is available" experiment, we weight the loss on each output index $a \in \{0, \ldots, 9\}$ according to $1/(a+1)^2$ and weight the help action by one. See Table 1 for the batch sizes and the number of epochs run in each MNIST experiment.

For all MNIST experiments, we used an NVIDIA Tesla V100 GPU and a 2.2 GHz Intel® Xeon® CPU with 100 GB memory. Since we use a neural network ensemble with $2,000$ models for each experiment, it takes about 50 GPU hours for each experiment, which makes a total of 300 GPU hours for all of our MNIST experiments.

---

**Algorithm 2** Learning to Be Cautious

---

Initialize $N$, $k$, and number of iterations $T$

Train $N * T$ reward functions $R(r|X)$ using Deep ensemble models on familiar MDP

**Input:** dataset $X$ containing images (e.g., MNIST, Fashion-MNIST, EMNIST)

Initialize: Total Regret $P^T = 0$, a random policy $\pi_0(a|X)$

**for** $t = 0$ **to** $T$ **do**

    Sample n reward functions $R(r|X)$ from the ensemble

    Inference rewards from n reward functions $R_n$

    Estimated state value:

$$V_\pi(X) = \sum_{i=0}^{m} R_n(r|x_i) * \pi_t(a|x_i)$$

    Select least $k$ estimated state value $V_\pi(X)$ and corresponding $k$ least reward functions. and Calculate mean of $k$ least reward functions

$$W_\mu = \frac{1}{k} \sum_{j=0}^{k} R_j(r|X)$$

    Immediate regret:

$$P^t = W_\mu - \sum_{a=0}^{|A|} \pi_t(a|X) * W_\mu$$

    Total Regret: $P^T = P^T + \max(0, P^t)$

    Update policy: $\pi_{t+1}(a|X) = \frac{P^T}{\sum_{a=0}^{|A|} P^T}$

  **end for**

---

Table 1: The batch size and number of epochs used to train the neural network reward models for each setting in the "how caution depends on the extent of training data" experiment. All other MNIST experiments use the same settings as in the 100% case.

| training data fraction | batch size | # of epochs |
|---|---|---|
| 1% | 64 | $10,000$ |
| 10% | 128 | $1,000$ |
| 100% | 512 | 100 |

The code for each MNIST experiment is available in a zip file The `actions` directory contains $k$-of-$N$ policies and expected values across each iteration and run in the MNIST experiments. The MNIST experiment code can be found in `code/mnist_experiments_code` split into a single shared file `k_of_n.py` and two Jupyter notebooks for each experiment, one to train neural networks and run $k$-of-$N$ CFR, and another to generate plots. The driving gridworld code can be found in `code/driving_gridworld_experiment_code`.

### B.1.1 K-of-N convergence

The progress of each $k$-of-$N$ CFR policy, measured in terms of expected return on the $N$ reward functions used on each CFR iteration, is given for each MNIST experiment in Figures 7, 10, 13 and 17 to 19. The progress of $k$-of-$N$ CFR in the driving gridworld is similar. The value always plateaus relatively quickly with little variation between runs, indicating that running more iterations or more repetitions would not change the results substantially. Because each run uses different sets of $N$ reward functions on each iteration by design, the value would still show some variation even if the policies for different runs were identical.

### B.1.2 Confusion matrix

We analyzed the behavior of $k$-of-$N$ CFR policies on the E-MNIST dataset by examining the confusion matrix of the final action probabilities for each letter class. The matrix reveals that, in the "learning to ask for help" experiment, for most letters, the policies confidently select the classification action rather than the "help" action, indicating generalization from previously monitored digit classes. However, a distinct pattern emerges for specific letters that visually resemble digits. For instance, the policies are more likely to select the "help" action when shown the letters o, s, i, l, j, and z, which closely resemble the digits 0, 5, 1, and 2, respectively, as shown in Figure 9. This behavior suggests that the agent recognizes higher epistemic uncertainty in such ambiguous cases and opts for monitoring to avoid potential misclassification.

We analyzed the behavior of $k$-of-$N$ CFR policies on the E-MNIST dataset by examining the confusion matrix of the final action probabilities for each letter class. The heatmaps for these confusion matrices are presented in Figure 12 for the "discovering non-obvious cautious actions" task, and in Figure 16 for the "ask for Help Only When It Is Available" task. Meanwhile, the confusion matrices for the "how caution depends on the extent of training data" task are shown in Figures 24 and 25.

### B.1.3 "last action" reward sensitivity

We conducted a sensitivity analysis on the reward associated with the "help" action in both the "learning to ask for help" and "ask for help only when it is available" tasks.

In the "learning to ask for help" experiment, we varied the "help" reward across $0.1, 0.25, 0.5$ in both the Fashion and E-MNIST environments. The results show that when the reward is low (e.g., 0.1), cautious agents tend to request help less frequently. In contrast, with higher rewards (e.g., 0.5), help is selected almost 100% of the time. Importantly, the relative performance ranking among cautious policies remains consistent across these values, as shown in Figure 5.

We evaluated accuracy and the frequency of selecting the "help" action across varying levels of the "help" reward ($\in 0.1, 0.25, 0.5$). Across all reward values, the "all-images" regime maintained consistently high classification accuracy ($> 99\%$), with a negligible reliance on help, even at the highest reward level (0.5); help was selected less than 1% of the time. In contrast, the "single-image" regime showed greater variability. At low help rewards (0.1), agents rarely used the help action, while at higher rewards (e.g., 0.5), the frequency of requesting help increased substantially (up to 7.63% in the 1-of-20 case), particularly under more cautious settings. Importantly, as the "help" reward increased, "help" usage rose and accuracy improved, suggesting that agents learned to request help when beneficial, especially under epistemic uncertainty. These findings validate that our cautious policies adapt help-seeking behavior based on its utility, while maintaining strong performance across regimes, as shown in Tables 2 to 4

Table 2: Frequency of the correct label action index and the help action across the 10,000 MNIST test images in the "Learning to Ask for Help" task, when the "help" reward = 0.1. Values reported as 0.00 are not necessarily zero but have been rounded to two decimal places.

|  |  | 1-of-20 | 1-of-10 | 5-of-10 | 10-of-10 |
|---|---|---|---|---|---|
| all-images regime | Correct | 99.38±0.03 | 99.38±0.02 | 99.39±0.01 | 99.40±0.01 |
|  | Help | 0.00±0.00 | 0.00±0.00 | 0.00±0.00 | 0.00±0.00 |
| single-image regime | Correct | 98.83±0.11 | 99.16±0.06 | 99.37±0.04 | 99.39±0.01 |
|  | Help | 0.00±0.00 | 0.00±0.00 | 0.00±0.00 | 0.00±0.00 |

In the "ask for help only when it is available" experiment, we evaluated rewards in $0.02, 0.05, 0.2$ to understand behavior when help availability is intermittent. As expected, higher "help" rewards increased the frequency of selecting the help action when it was available, as illustrated in Figure 6.

The most cautious policy (1-of-20) in the all-images regime selects the help action upon observing most letters except for those similar to digits (*e.g.*, I/i, L/l, O/o, S/s and Z/z) as shown in Figure 9. The effect

Table 3: Frequency of the correct label action index and the help action across the 10,000 MNIST test images in the "Learning to Ask for Help" task, when the "help" reward = 0.25

|  |  | 1-of-20 | 1-of-10 | 5-of-10 | 10-of-10 |
|---|---|---|---|---|---|
| all-images regime | Correct | 99.35±0.02 | 99.36±0.03 | 99.38±0.01 | 99.39±0.01 |
|  | Help | 0.03±0.01 | 0.03±0.01 | 0.02±0.00 | 0.02±0.00 |
| single-image regime | Correct | 96.87±0.05 | 97.95±0.07 | 99.07±0.02 | 99.39±0.01 |
|  | Help | 2.83±0.01 | 1.72±0.04 | 0.41±0.01 | 0.02±0.00 |

Table 4: Frequency of the correct label action index and the help action across the 10,000 MNIST test images in the "Learning to Ask for Help" task, when the "help" reward = 0.5

|  |  | 1-of-20 | 1-of-10 | 5-of-10 | 10-of-10 |
|---|---|---|---|---|---|
| all-images regime | Correct | 98.82±0.04 | 98.92±0.04 | 99.05±0.01 | 99.10±0.01 |
|  | Help | 0.89±0.04 | 0.76±0.05 | 0.57±0.01 | 0.52±0.01 |
| single-image regime | Correct | 92.37±0.12 | 94.94±0.08 | 99.08±0.02 | 99.10±0.01 |
|  | Help | 7.63±0.12 | 5.05±0.08 | 1.80±0.02 | 0.52±0.02 |

is exaggerated in the single-image regime where 1-of-20 selects the help action with a very high probability except for letters similar to digits. The baseline $\text{Optim}(\hat{r})$ does not select the help action at all.

The most cautious policy (1-of-20) in the all-images regime picks the help action when help is available and otherwise selects less risky actions with small indices upon observing most letters except for those similar to digits, as shown in Figure 16. The effect is exaggerated in the single-image regime.

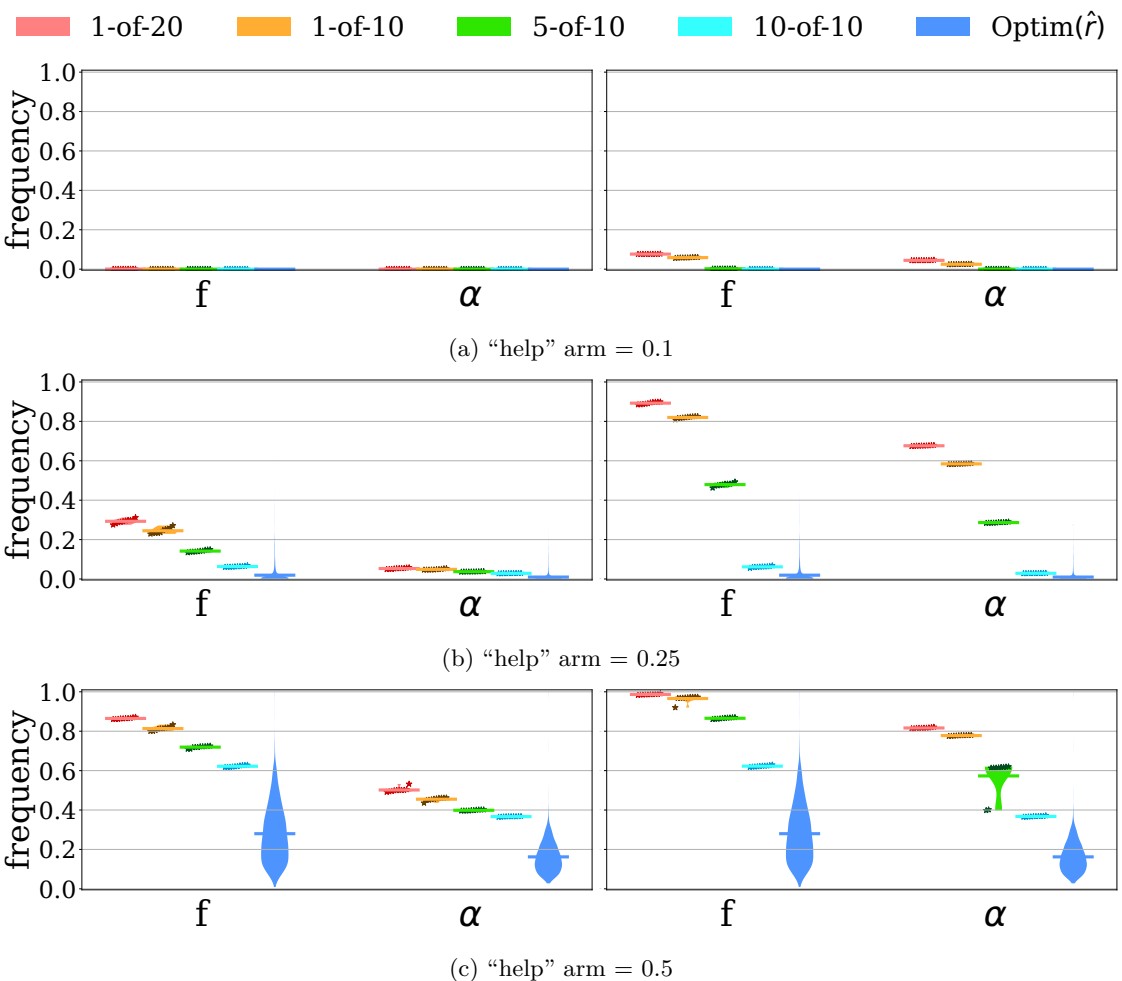

(a) "help" arm = 0.1

(b) "help" arm = 0.25

(c) "help" arm = 0.5

Figure 5: "Help" arm reward sensitivity in the "learning to ask for help" task, where the "help" reward $\in \{0.1, 0.25, 0.5\}$, ("f" for fashion and "$\alpha$" for letters). Each star represents a single run, and the violin represents the variance across 10 runs.

Table 5: Frequency of the correct label action index and the help action across the 10,000 MNIST test images in the "Learning to Ask for Help" task

|  |  | 1-of-20 | 1-of-10 | 5-of-10 | 10-of-10 |
|---|---|---|---|---|---|
| all-images regime | Correct | 99.35±0.02 | 99.36±0.03 | 99.38±0.01 | 99.39±0.01 |
|  | Help | 0.03±0.01 | 0.03±0.01 | 0.02±0.00 | 0.02±0.00 |
| single-image regime | Correct | 96.87±0.05 | 97.95±0.07 | 99.07±0.02 | 99.39±0.01 |
|  | Help | 2.83±0.01 | 1.72±0.04 | 0.41±0.01 | 0.02±0.00 |

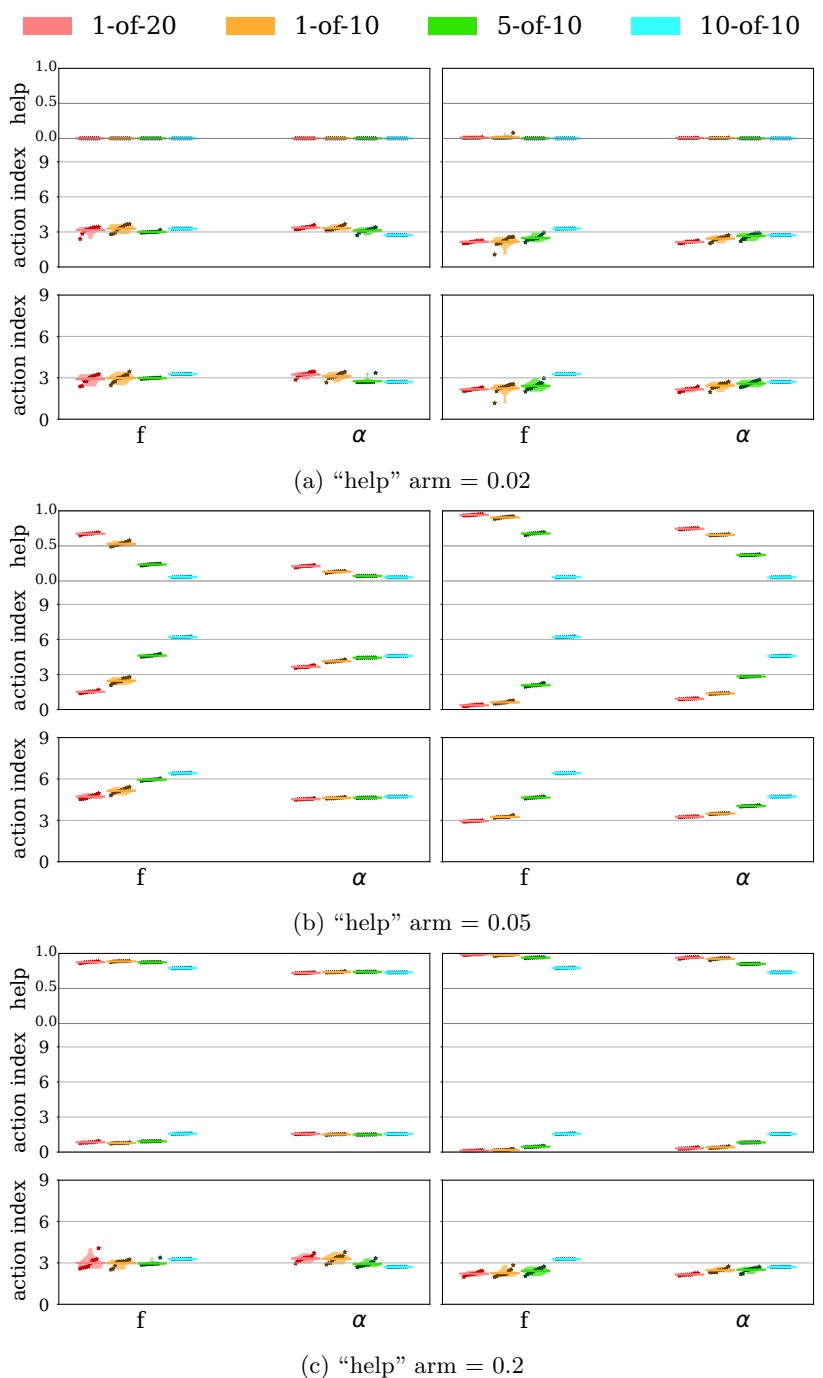

Figure 6: Average action index and frequency of selecting the "help" action by each method in various novel environments, under different rewards for the "help" arm in the "learning to ask for help only when available" task. The "help" reward takes values $\in \{0.02, 0.05, 0.2\}$, ("f" for fashion and "$\alpha$" for letters). Each star represents a single run, and the violin represents the variance across 10 runs.

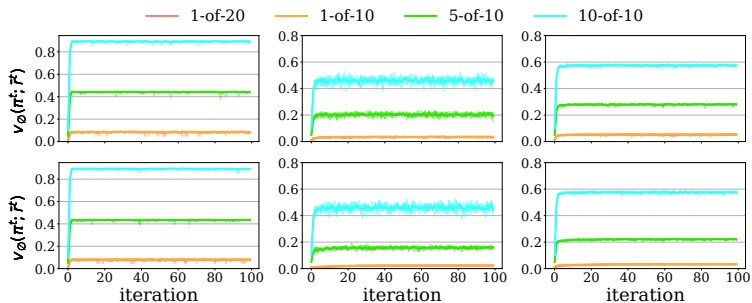

Figure 7: The expected return of each $k$-of-$N$ policy on each iteration $t$ given the sampled $k$-of-$N$ reward function, $\bar{r}^t$, in the "learning to ask for help" task. A single bold line shows the average across all ten runs while the values from individual runs are given by thinner lines. (top row) All-images regime, (bottom row) single-image regime, (left column) digits, (middle column) fashion, (right column) letter.

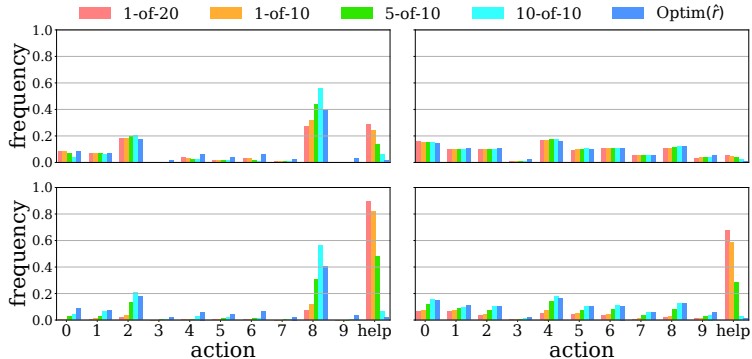

Figure 8: Action distribution of each $k$-of-$N$ policy and baseline in the "learning to ask for help" task. (top row) All-images regime, (bottom row) single-image regime, (left column) fashion, (right column) letter.

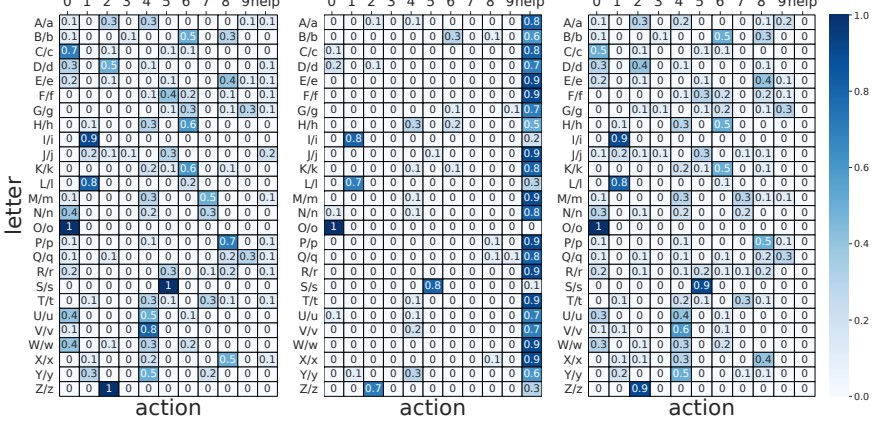

Figure 9: The average frequency of each action on each letter, averaged over lowercase and uppercase images, in the "learning to ask for help" task. (left) 1-of-20 all-images regime, (middle) 1-of-20 single-image regime, (right) Optim($\hat{r}$).

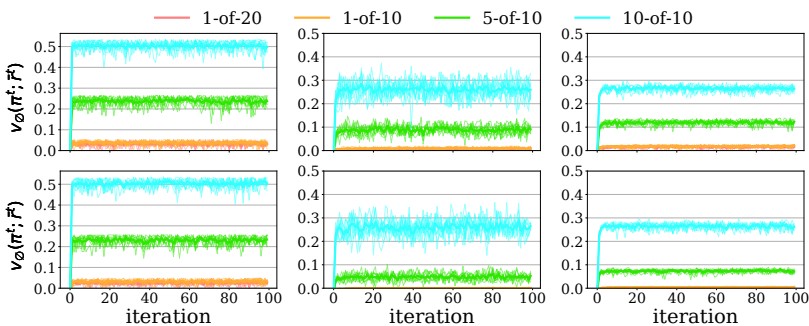

Figure 10: The expected return of each $k$-of-$N$ policy on each iteration $t$ given the sampled $k$-of-$N$ reward function, $\bar{r}^t$, in the "discovering non-obvious cautious actions" task. (top row) All-images regime, (bottom row) single-image regime, (left column) digits, (middle column) fashion, (right column) letter.

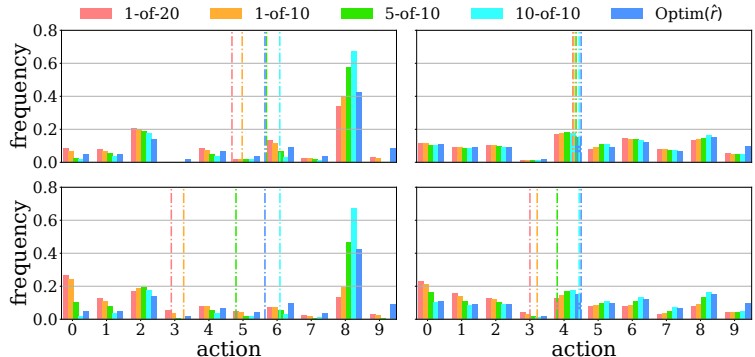

Figure 11: Action distribution of each $k$-of-$N$ policy and baseline in the "discovering non-obvious cautious actions" task, dotted lines represent average action taken by each policy. (top row) All-images regime, (bottom row) single-image regime, (left column) fashion, (right column) letter.

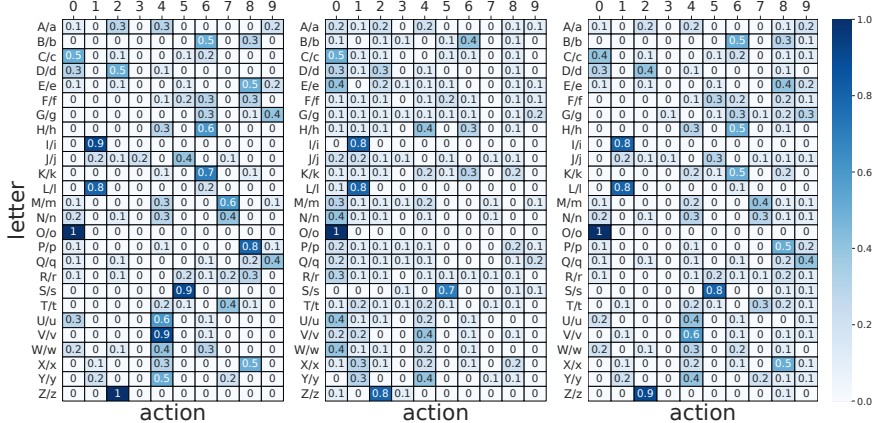

Figure 12: The average frequency of each action on each letter, averaged over lowercase and uppercase images, in the "discovering non-obvious cautious actions" task. (left) 1-of-20 all-images regime, (middle) 1-of-20 single-image regime, (right) Optim($\hat{r}$).

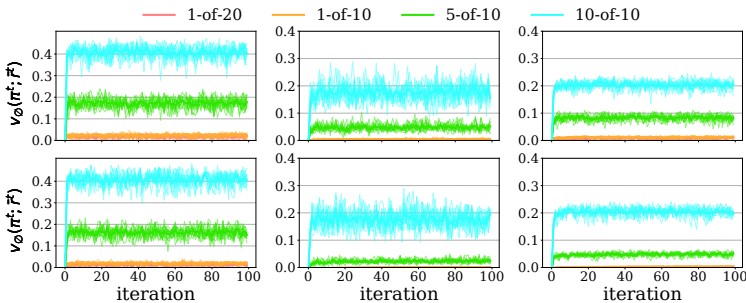

Figure 13: The expected return of each $k$-of-$N$ policy on each iteration $t$ given the sampled $k$-of-$N$ reward function, $\bar{r}^t$, in the "ask for help only when it is available" task. (top row) All-images regime, (bottom row) single-image regime, (left column) digits, (middle column) fashion, (right column) letter.

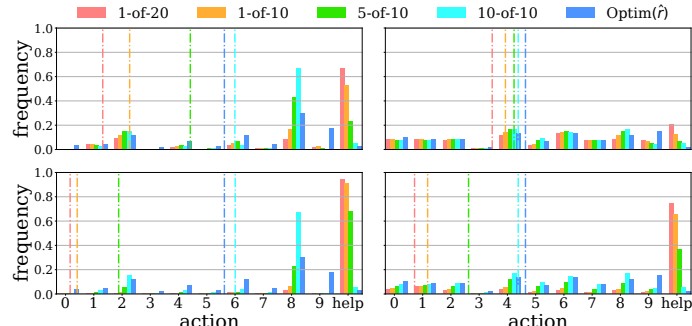

Figure 14: Actions distribution for each $k$-of-$N$ policy and baseline in the "ask for help only when it is available" task in case that **help is available**. dotted lines represent the average action taken by each policy. (top row) All-images regime, (bottom row) single-image regime, (left column) fashion, (right column) letter.

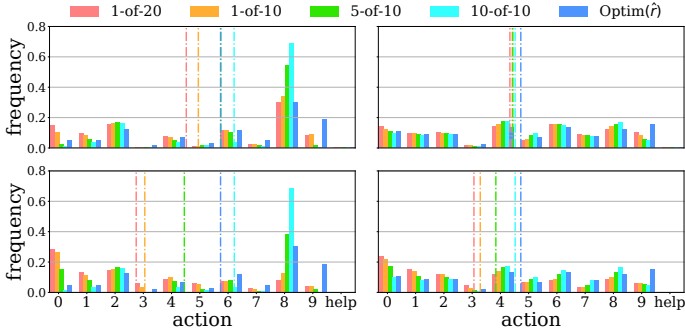

Figure 15: Actions distribution for each $k$-of-$N$ policy and baseline in the "ask for help only when it is available" task in case that **help is unavailable**. dotted lines represent the average action taken by each policy. (top row) All-images regime, (bottom row) single-image regime, (left column) fashion, (right column) letter.

Table 6: Frequency of the correct label action index and the help action across the 10,000 MNIST test images in the "ask for help only when it is available" all-images regime.

|  |  | 1-of-20 | 1-of-10 | 5-of-10 | 10-of-10 | Optim($\hat{r}$) |
|---|---|---|---|---|---|---|
| help is available | Correct | 97.50±3.11 | 98.57±1.00 | 99.27±0.01 | 99.27±0.01 | 89.43±4.16 |
|  | Help | 0.39±0.48 | 0.14±0.02 | 0.09±0.00 | 0.07±0.00 | 1.16±0.00 |
| help is unavailable | Correct | 97.64±3.05 | 98.67±0.99 | 99.32±0.01 | 99.31±0.01 | 94.96±4.01 |
|  | Help | 0.00±0.00 | 0.00±0.00 | 0.00±0.00 | 0.00±0.00 | 0.23±3.87 |

Table 7: Frequency of the correct label action index and the help action across the 10,000 MNIST test images in the "ask for help only when it is available" single-image regime.

| | | 1-of-20 | 1-of-10 | 5-of-10 | 10-of-10 | Optim($\hat{r}$) |
|---|---|---|---|---|---|---|
| help is available | Correct | 84.13±1.07 | 90.68±1.54 | 98.80±0.01 | 99.28±0.01 | 89.43±4.16 |
| | Help | 10.42±0.51 | 3.25±0.47 | 0.05±0.01 | 0.07±0.01 | 1.16±0.00 |
| help is unavailable | Correct | 88.54±1.14 | 92.18±1.59 | 99.06±0.02 | 99.31±0.01 | 94.96±4.01 |
| | Help | 0.00±0.00 | 0.00±0.00 | 0.00±0.00 | 0.00±0.00 | 0.23±3.87 |

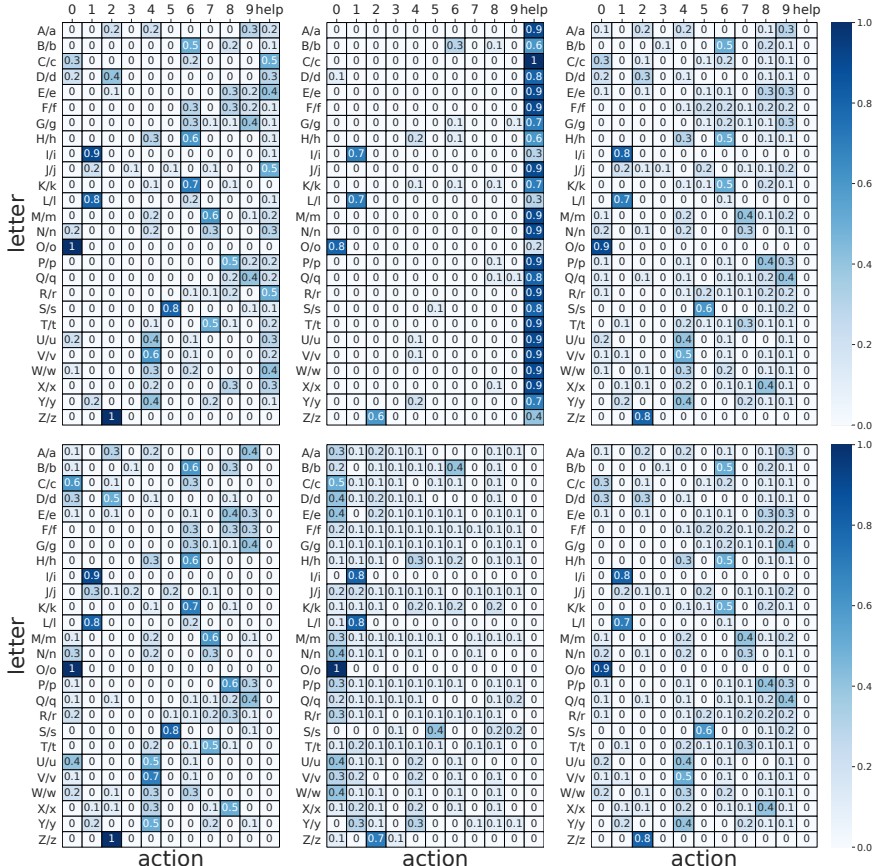

Figure 16: The average frequency of each action on each letter, averaged over lowercase and uppercase images, in the "ask for help only when it is available" task. (top row) Help is available (bottom row), help is unavailable. (Left column) 1-of-20 all-images regime, (middle column) 1-of-20 single-image regime, (right column) Optim($\hat{r}$).

Table 8: Frequency of the correct label action index and the help action across the 10,000 MNIST test images in the "learning to ask for help" task with perturbed rewards in the all-images regime, where reward models are trained on 1%, 10%, or 100% of the digit dataset.

|  |  | 1-of-20 | 1-of-10 | 5-of-10 | 10-of-10 | Optim($\hat{r}$) |
|---|---|---|---|---|---|---|
| 1% | Correct | 95.81±0.14 | 95.82±0.18 | 95.97±0.08 | 96.10±0.06 | 89.43±4.16 |
|  | Help | 0.39±0.03 | 0.41±0.07 | 0.34±0.02 | 0.30±0.03 | 1.16±0.93 |
| 10% | Correct | 98.63±0.07 | 98.62±0.08 | 98.67±0.04 | 98.67±0.03 | 94.96±4.01 |
|  | Help | 0.13±0.01 | 0.12±0.01 | 0.12±0.01 | 0.10±0.01 | 0.23±3.87 |
| 100% | Correct | 99.34±0.04 | 99.36±0.04 | 99.38±0.02 | 99.38±0.02 | 98.23±3.57 |
|  | Help | 0.04±0.00 | 0.03±0.01 | 0.03±0.00 | 0.02±0.00 | 0.08±0.73 |

Table 9: Frequency of the correct label action index and the help action across the 10,000 MNIST test images in the "learning to ask for help" task with perturbed rewards in the single-image regime, where reward models are trained on 1%, 10%, or 100% of the digit dataset.

|  |  | 1-of-20 | 1-of-10 | 5-of-10 | 10-of-10 | Optim($\hat{r}$) |
|---|---|---|---|---|---|---|
| 1% | Correct | 86.96±0.41 | 89.75±0.34 | 94.56±0.12 | 96.11±0.06 | 89.43±4.16 |
|  | Help | 5.80±0.26 | 4.49±0.26 | 1.61±0.06 | 0.29±0.02 | 1.16±0.93 |
| 10% | Correct | 88.78±0.16 | 92.51±0.13 | 97.50±0.06 | 98.68±0.03 | 94.96±4.0 |
|  | Help | 9.08±0.10 | 6.61±0.09 | 1.16±0.02 | 0.11±0.01 | 0.23±3.87 |
| 100% | Correct | 96.61±0.07 | 97.67±0.06 | 99.07±0.03 | 99.37±0.02 | 98.23±3.57 |
|  | Help | 2.47±0.02 | 1.52±0.03 | 0.36±0.01 | 0.02±0.00 | 0.08±0.73 |

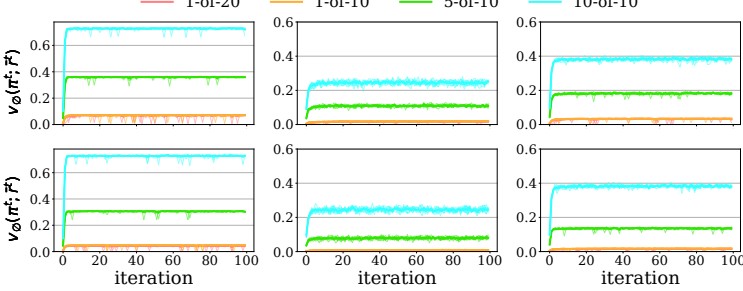

Figure 17: The expected return of each $k$-of-$N$ policy on each iteration $t$ given the sampled $k$-of-$N$ reward function, $\bar{r}^t$, in the "learning to ask for help" task with perturbed rewards, where reward models are trained on 1% of the digit dataset. (top row) All-images regime, (bottom row) single-image regime, (left column) digits, (middle column) fashion, (right column) letter.

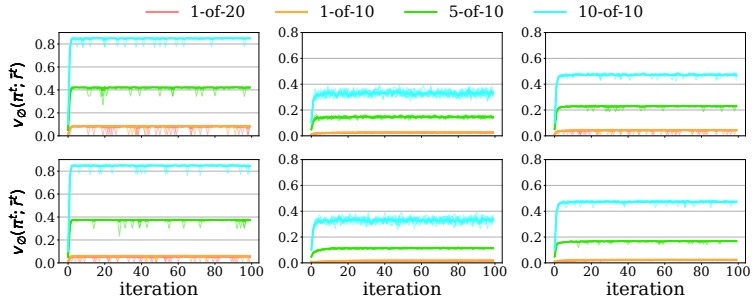

Figure 18: The expected return of each $k$-of-$N$ policy on each iteration $t$ given the sampled $k$-of-$N$ reward function, $\bar{r}^t$, in the "learning to ask for help" task with perturbed rewards, where reward models are trained on 10% of the digit dataset. (top row) All-images regime, (bottom row) single-image regime, (left column) digits, (middle column) fashion, (right column) letter.

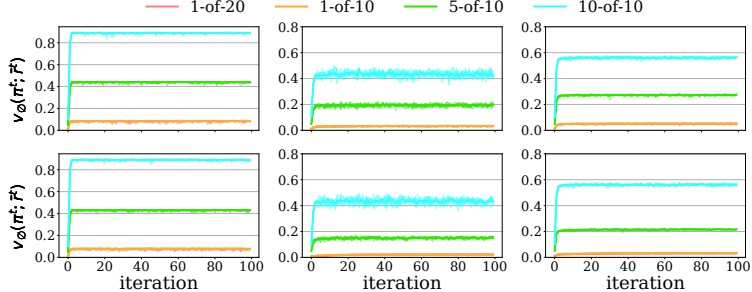

Figure 19: The expected return of each $k$-of-$N$ policy on each iteration $t$ given the sampled $k$-of-$N$ reward function, $\bar{r}^t$, in the "learning to ask for help" task with perturbed rewards, where reward models are trained on 100% of the digit dataset. (top row) All-images regime, (bottom row) single-image regime, (left column) digits, (middle column) fashion, (right column) letter.

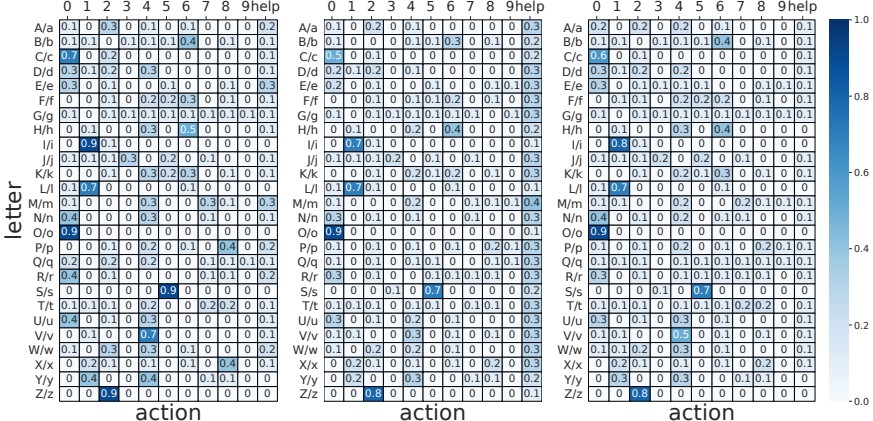

Figure 20: The average frequency of each action on each letter, averaged over lowercase and uppercase images, in the "learning to ask for help" task with perturbed rewards, where reward models are trained on 1% of the digit dataset. (left) 1-of-20 all-images regime, (middle) 1-of-20 single-image regime, (right) Optim($\hat{r}$).

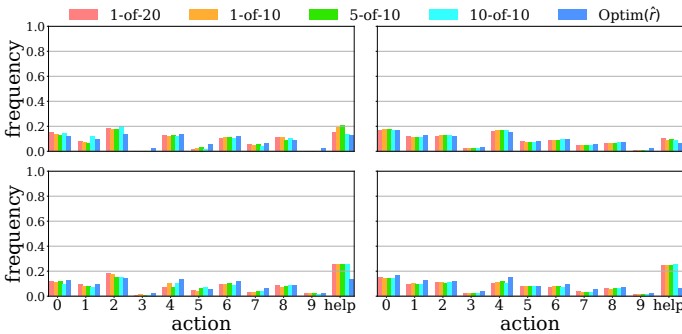

Figure 21: Action distribution of each $k$-of-$N$ policy and baseline in the "learning to ask for help" task with perturbed rewards, where reward models are trained on 1% of the digit dataset. (top row) All-images regime, (bottom row) single-image regime, (left column) fashion, (right column) letter.

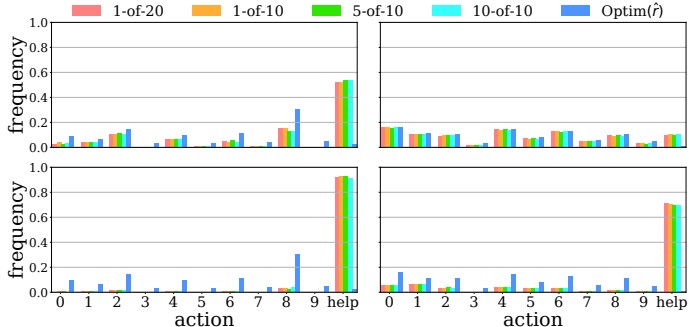

Figure 22: Action distribution of each $k$-of-$N$ policy and baseline in the "learning to ask for help" task with perturbed rewards, where reward models are trained on 10% of the digit dataset. (top row) All-images regime, (bottom row) single-image regime, (left column) fashion, (right column) letter.

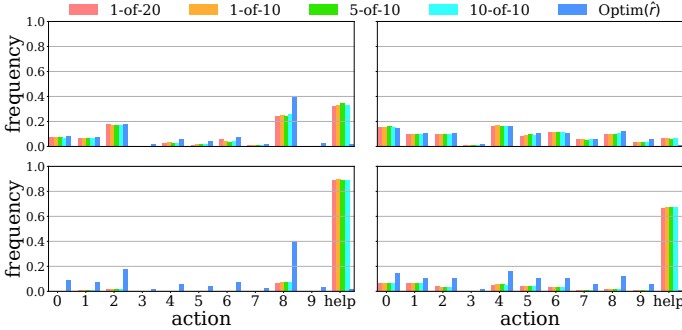

Figure 23: Action distribution of each $k$-of-$N$ policy and baseline in the "learning to ask for help" task with perturbed rewards, where reward models are trained on 100% of the digit dataset. (top row) All-images regime, (bottom row) single-image regime, (left column) fashion, (right column) letter.

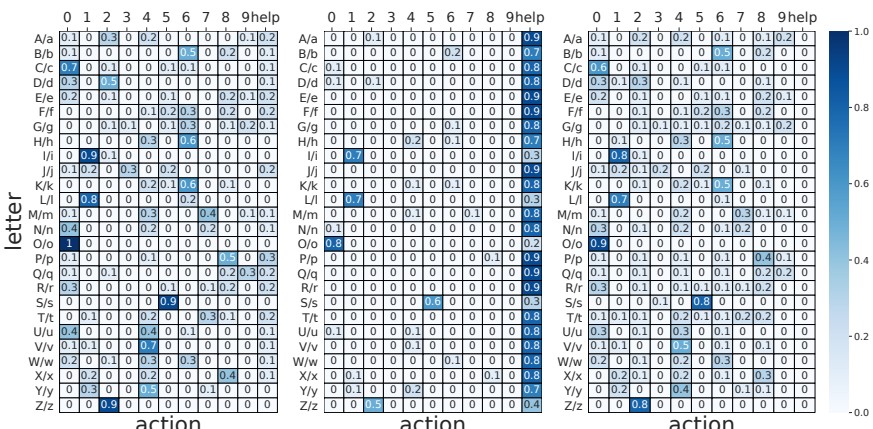

Figure 24: The average frequency of each action on each letter, averaged over lowercase and uppercase images, in the "learning to ask for help" task with perturbed rewards, where reward models are trained on 10% of the digit dataset. (left) 1-of-20 all-images regime, (middle) 1-of-20 single-image regime, (right) Optim($\hat{r}$).

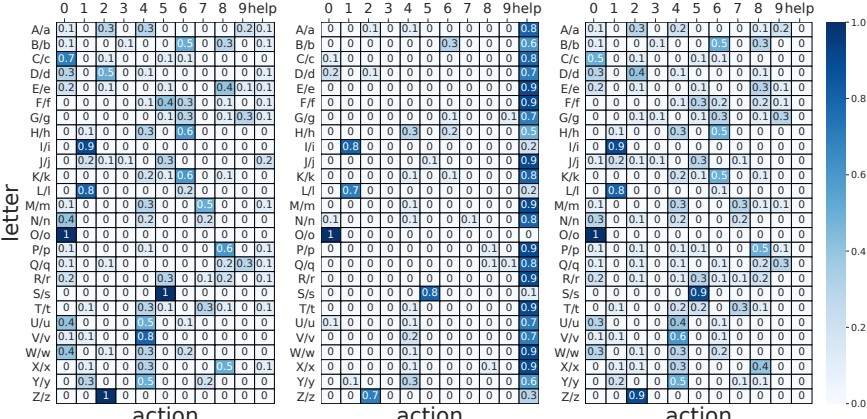

Figure 25: The average frequency of each action on each letter, averaged over lowercase and uppercase images, in the "learning to ask for help" task with perturbed rewards, where reward models are trained on 100% of the digit dataset. (left) 1-of-20 all-images regime, (middle) 1-of-20 single-image regime, (right) Optim($\hat{r}$).

### B.2 The Driving Gridworld Experiment

For the driving gridworld experiment, the training data for our neural networks are generated by enumerating all of the (state, next state, reward)-tuples in the familiar driving gridworld where obstacles can only appear in either of the two ditch lanes on the far left or far right column. The action is omitted because the familiar driving gridworld's reward function depends only on the initial and next state. Each driving gridworld state image is pre-processed into a four-channel image where each channel encodes the positions of different aspects of the environment: pavement, ditch, car, and obstacle.

Our networks have two parallel convolutional layers with four output channels and $2 \times 2$ filters, each followed by a ReLU transformation. The outputs from these layers are flattened and concatenated together, and that result is concatenated with a one-hot encoding of the car's speed in the next state. Next, we apply two fully connected layers separated by a ReLU function, the first with 32 outputs and the second with a single output, respectively. For each possible speed the car could have in the initial state, we manage a different pair of fully connected layers with the same shapes. To compute the expected reward for a given action in a given state, we query the neural network with every possible state that could follow from the given action and use the transition probabilities to combine these state–next state reward estimates.

Networks are trained over $51,200$ epochs using a batch size of 800 to minimize the MSE using Adam with a learning rate of 0.0001 and weight decay of $10^{-5}$. The remaining parameters for Adam in PyTorch ($\beta_1$, $\beta_2$, and $\epsilon$) are set to their defaults ($0.9$, $0.999$, $10^{-8}$) without the AMSGrad (Reddi et al., 2018) modification.

Policies are evaluated by iterative dynamic programming according to the Bellman operator. An approximation of the $\gamma$-discounted expected return function is updated simultaneously for each state and action until the maximum absolute change is smaller than $10^{-6}$. We use a discount factor of $\gamma = 0.99$. On every $k$-of-$N$ CFR iteration, this process is run $N$ times given the current CFR policy to evaluate it under each of the $N$ reward functions sampled at the start of the iteration[8].

This experiment was run on a 3.60GHz Intel® Core™ i9-9900K CPU with 7.7 GB of memory without a GPU. Model training took roughly 25 minutes per random initialization, so 833 CPU hours for all $2,000$ models. 100 iterations of $k$-of-$N$ CFR took roughly three minutes, so for all six settings of $k$ and $N$, and all five random repetitions, it took about 100 minutes. In total, our experiment used about 835 CPU hours of computation.

---

[8]Alternatively, the successor representation (Dayan, 1993) could be used to fully characterize the current CFR policy for essentially the same cost as a single policy evaluation. Given this information, computing the expected return given a reward function can be computed with a single pass over each state and action, thereby reducing the amount of computation that scales with $N$.

## C  $k$-of-$N$ CFR for Continuing MDPs

Here we restate and prove our theoretical results for CFR and $k$-of-$N$ CFR in continuing MDPs with reward uncertainty.

Define

$$d_s : s'; \pi \mapsto (1-\gamma)\mathbb{E}\left[\sum_{i=0}^{\infty} \gamma^i \mathbb{1}\{S_i = s'\} \,|\, S_0 = s\right],$$

where $A_i \sim \pi(\cdot|S_{i-1})$ and $S_i \sim p(\cdot|S_{i-1}, A_i)$ for $i \geq 1$, to be the $\gamma$-discounted future state distribution induced by $\pi$ from initial state $s$. Kakade (2003)'s performance difference lemma for this setting is:

**Lemma 1.** *The full regret for using stationary policy $\pi$ instead of stationary competitor policy $\pi'$ from state $s$ in MDP $(\mathcal{S}, \mathcal{A}, p, d_\varnothing, \gamma)$ under reward function $r$ is*

$$v_s(\pi'; r) - v_s(\pi; r) = \frac{1}{1-\gamma}\mathbb{E}[\rho_S(A, \pi; r)], \ \text{where } S \sim d_s(\cdot; \pi') \text{ and } A \sim \pi'(\cdot|S).$$

**Theorem 1.** *CFR bounds cumulative full regret with respect to any stationary policy $\pi$ as*

$$\sum_{t=1}^{T} v_\varnothing(\pi; r^t) - v_\varnothing(\pi^t; r^t) \leq C^T/(1-\gamma).$$

*Proof.* Let $S \sim d_s(\cdot; \pi)$ and $A \sim \pi(\cdot|S)$. By Lemma 1, the linearity of expectation, and CFR's definition,

$$\sum_{t=1}^{T} v_s(\pi; r^t) - v_s(\pi^t; r^t) = \frac{1}{1-\gamma}\mathbb{E}\left[\sum_{t=1}^{T} \rho_S(A, \pi^t; r^t)\right] \tag{1}$$

$$\leq \frac{1}{1-\gamma}\mathbb{E}[C^T] \tag{2}$$

$$= C^T/(1-\gamma), \tag{3}$$

for each state $s$. Since this bound holds for all states simultaneously, it holds for the $\gamma$-discounted expected return as long as the initial state distribution is constant across iterations, *i.e.*, assuming that the initial state is $S \sim d_\varnothing$ on each iteration,

$$\sum_{t=1}^{T} v_\varnothing(\pi; r^t) - v_\varnothing(\pi^t; r^t) = \sum_{t=1}^{T} E[v_S(\pi; r^t) - v_S(\pi^t; r^t)] \tag{4}$$

$$= E\left[\sum_{t=1}^{T} v_S(\pi; r^t) - v_S(\pi^t; r^t)\right] \tag{5}$$

$$\leq E[C^T/(1-\gamma)] \tag{6}$$

$$= C^T/(1-\gamma), \tag{7}$$

which completes the argument. □

We make use of the Azuma-Hoeffding inequality in the $k$-of-$N$ CFR regret bound so it is restated here for completeness:

**Proposition 1** (Azuma-Hoeffding inequality). *For constants $(c^t)_{t=1}^T$, martingale difference sequence $(Y^t)_{t=1}^T$ where $|Y^t| \leq c^t$ for each $t$, and $\tau \geq 0$,*

$$\mathbb{P}\left[\left|\sum_{t=1}^{T} Y^t\right| \geq \tau\right] \leq 2\exp\frac{-\tau^2}{2\sum_{t=1}^{T}(c^t)^2}.$$

For proof, see that of Theorem 3.14 by McDiarmid (1998).

We construct the following regret bound for $k$-of-$N$ CFR based on Theorem 1:

**Theorem 2.** *With probability $1 - p$, $p > 0$, the full regret of $k$-of-N CFR with respect to any stationary policy, $\pi$, is upper bounded by*

$$\frac{C^T}{1 - \gamma} + 4U\sqrt{2T \log 1/p}.$$

*Proof.* Let the reward function distribution be $\mathcal{R}$. On each iteration, $t$, of the $k$-of-$N$ CFR algorithm, $N$ reward functions, $(r_j^t \sim \mathcal{R})_{j=1}^N$ are sampled and the worst $k$ reward functions for the algorithm's current policy, $\pi^t$, are mixed into $\bar{r}^t : s, a, s' \mapsto \frac{1}{k} \sum_{j=1}^k r_{(j)}^t(s, a, s')$ where $r_{(j)}^t$ is the $j^{\text{th}}$-worst reward function. The $k$-element average reward function $\bar{r}^t$ is a sample from $\mathcal{R}$ according to the $k$-of-$N$ probability measure, $\mu_{k\text{-of-}N}$ (see Proposition 1 of Chen & Bowling (2012) for a formal description of this measure's density) where the reward functions are ranked with respect to $\pi^t$.

Let $\mu_{k\text{-of-}N}(\cdot; \pi^t)$ denote the $k$-of-$N$ reward function distribution with respect to policy $\pi^t$. The expected return of $\pi^t$ under the $k$-of-$N$ robustness objective is then just the expectation of its return under $\bar{r}^t \sim \mu_{k\text{-of-}N}(\cdot; \pi^t)$, *i.e.*, $\mathbb{E}[v_\varnothing(\pi^t; \bar{r}^t)]$. Thus, we seek to bound the regret under this payoff function,

$$\sum_{t=1}^T \rho_\varnothing(\pi; \pi^t) \doteq \sum_{t=1}^T \mathbb{E}[v_\varnothing(\pi; \bar{r})] - \mathbb{E}[v_\varnothing(\pi^t; \bar{r}^t)] \tag{8}$$

$$\leq \sum_{t=1}^T \mathbb{E}[v_\varnothing(\pi; \bar{r}^t)] - \mathbb{E}[v_\varnothing(\pi^t; \bar{r}^t)] \tag{9}$$

$$= \sum_{t=1}^T \mathbb{E}[\underbrace{v_\varnothing(\pi; \bar{r}^t) - v_\varnothing(\pi^t; \bar{r}^t)}_{\doteq \bar{\rho}_\varnothing(\pi; \pi^t)}], \tag{10}$$

where $\bar{r} \sim \mu_{k\text{-of-}N}(\cdot; \pi)$ and where the inequality follows from the fact that ranking the reward functions according to $\pi^t$ instead of $\pi$ can only improve $\pi$'s return. The $\bar{\rho}_\varnothing(\pi; \pi^t)$ terms in the sum and expectation are the random instantaneous regrets that the $k$-of-$N$ CFR algorithm computes on each iteration.

The rest of the proof largely follows the proof of Farina et al. (2020)'s Proposition 1. Since $\mathbb{E}[\bar{\rho}_\varnothing(\pi; \pi^t)]$ is the expectation of $\bar{\rho}_\varnothing(\pi; \pi^t)$, the sequence of differences,

$$\left(\mathbb{E}[\bar{\rho}_\varnothing(\pi; \pi^t)] - \bar{\rho}_\varnothing(\pi; \pi^t)\right)_{t=1}^T,$$

is a martingale difference sequence. Furthermore, $|\mathbb{E}[\bar{\rho}_\varnothing(\pi; \pi^t)] - \bar{\rho}_\varnothing(\pi; \pi^t)| \leq 4U$ since regret can only be twice as large as the largest return and returns are bounded by the largest reward (thanks to normalization).

The probability that the cumulative regret, $\sum_{t=1}^T \rho_\varnothing(\pi; \pi^t)$, is bounded by the cumulative sampled regret plus slack $\tau \geq 0$ is bounded according to the Azuma-Hoeffding inequality (Proposition 1):

$$\mathbb{P}\left[\sum_{t=1}^T \rho_\varnothing(\pi; \pi^t) \leq \sum_{t=1}^T \bar{\rho}_\varnothing(\pi; \pi^t) + \tau\right] \leq \mathbb{P}\left[\sum_{t=1}^T \mathbb{E}[\bar{\rho}_\varnothing(\pi; \pi^t)] - \bar{\rho}_\varnothing(\pi; \pi^t) \leq \tau\right] \tag{11}$$

$$= 1 - \mathbb{P}\left[\sum_{t=1}^T \mathbb{E}[\bar{\rho}_\varnothing(\pi; \pi^t)] - \bar{\rho}_\varnothing(\pi; \pi^t) \geq \tau\right] \tag{12}$$

$$\leq 1 - \exp \frac{2\tau^2}{4T(4U)^2}. \tag{13}$$

Setting $\tau = 4U\sqrt{2T \log(1/p)}$ ensures that

$$\sum_{t=1}^T \rho_\varnothing(\pi; \pi^t) \leq \sum_{t=1}^T \bar{\rho}_\varnothing(\pi; \pi^t) + 4U\sqrt{2T \log 1/p}$$

with probability $1 - p$. Since $\sum_{t=1}^{T} \bar{\rho}_{\varnothing}(\pi; \pi^t) \leq C^T/(1 - \gamma)$,

$$\sum_{t=1}^{T} \rho_{\varnothing}(\pi; \pi^t) \leq C^T/(1 - \gamma) + 4U\sqrt{2T \log 1/p}$$

with probability $1 - p$, as required. $\qquad\square$

The $k$-of-$N$ CFR optimality bound makes use of another elementary result, Markov's inequality:

**Proposition 2** (Markov's inequality). *The probability that non-negative random variable $X \geq 0$ is at least $a > 0$ is upper bounded by $X$'s expectation divided by $a$, i.e., $\mathbb{P}[X > a] \leq \mathbb{E}[X]/a$.*

*Proof.* By the law of total expectation,

$$\mathbb{E}[X] = \mathbb{E}[\mathbb{E}[X \mid X \leq a]] \tag{14}$$
$$= \mathbb{P}[X \leq a]\mathbb{E}[X \mid X \leq a] + \mathbb{P}[X > a]\mathbb{E}[X \mid X > a]. \tag{15}$$

Since $X$ is non-negative and that $\mathbb{E}[X \mid X > a]$ conditions on $X$ being no-smaller than $a$,

$$\geq \mathbb{P}[X \leq a]0 + \mathbb{P}[X > a]a \tag{16}$$
$$= \mathbb{P}[X > a]a. \tag{17}$$

Dividing both sides by $a$ yields the desired statement. $\qquad\square$

**Theorem 3.** *With probability $1 - p$, $p > 0$, the best policy in the sequence of policies generated by $k$-of-$N$ CFR, $(\pi^t)_{t=1}^{T}$, is an $\varepsilon^T$-approximation to a $\mu_{k\text{-of-}N}$-robust policy where*

$$\varepsilon^T = \frac{C^T}{(1 - \gamma)T} + 4U\sqrt{\frac{2 \log 1/p}{T}}$$

*and with probability at least $(1 - p)(1 - q)$, $q > 0$, a randomly sampled policy from this sequence is an $\varepsilon^T/q$-approximation to a $\mu_{k\text{-of-}N}$-robust policy.*

*Proof.* The first half of the proof is essentially that of Lockhart et al. (2019a;b)'s Lemma 2.

As before, let $\bar{r}^t \sim \mu_{k\text{-of-}N}(\cdot; \pi^t)$. Define $v_{\varnothing}^* = \max_{\pi^*} \mathbb{E}_{\bar{r} \sim \mu_{k\text{-of-}N}(\cdot; \pi^*)}[v_{\varnothing}(\pi^*; \bar{r})]$ to be the return of a $\mu_{k\text{-of-}N}$-robust policy, $\pi^*$. Since the competitor term of the regret does not depend on the iteration number, we can rewrite the average regret as

$$\varepsilon^T \geq \frac{1}{T}\sum_{t=1}^{T} v_{\varnothing}^* - \mathbb{E}[v_{\varnothing}(\pi^t; \bar{r}^t)] \tag{18}$$

$$= v_{\varnothing}^* - \frac{1}{T}\sum_{t=1}^{T} \mathbb{E}[v_{\varnothing}(\pi^t; \bar{r}^t)] \tag{19}$$

$$\geq v_{\varnothing}^* - \max_{\hat{\pi} \in (\pi^t)_{t=1}^{T}} \mathbb{E}_{\bar{r} \sim \mu_{k\text{-of-}N}(\cdot; \hat{\pi})}[v_{\varnothing}(\hat{\pi}; \bar{r})], \tag{20}$$

where the last inequality holds with probability $1 - p$ according to Theorem 2. Rearranging terms, we see that

$$\max_{\hat{\pi} \in (\pi^t)_{t=1}^{T}} \mathbb{E}_{\bar{r} \sim \mu_{k\text{-of-}N}(\cdot; \hat{\pi})}[v_{\varnothing}(\hat{\pi}; \bar{r})] \geq v_{\varnothing}^* - \varepsilon^T.$$

Thus, the best policy in the sequence, $\hat{\pi}$, achieves the optimal $k$-of-$N$ value minus $\varepsilon^T$ with probability $1 - p$, *i.e.*, $\hat{\pi}$ is an $\varepsilon^T$-approximation to a $\mu_{k\text{-of-}N}$-robust policy.

The last half of this proof is essentially that of Johanson et al. (2012)'s Theorem 4.

Let $\hat{\pi} \sim \mathrm{Unif}(\{\pi^t\}_{t=1}^T)$ be the random variable representing a uniformly sampled policy. Then,

$$\varepsilon^T \geq v_\varnothing^* - \frac{1}{T}\sum_{t=1}^T \mathbb{E}[v_\varnothing(\pi^t; \vec{r}^t)] \tag{21}$$

$$= \mathbb{E}\left[v_\varnothing^* - \mathbb{E}_{\bar{r}\sim\mu_{k\text{-of-}N}(\cdot;\hat{\pi})}[v_\varnothing(\hat{\pi}; \bar{r})]\right], \tag{22}$$

with probability $1 - p$ according to Theorem 2.

Let $X = v_\varnothing^* - \mathbb{E}_{\bar{r}\sim\mu_{k\text{-of-}N}(\cdot;\hat{\pi})}[v_\varnothing(\hat{\pi}; \bar{r})] \geq 0$. By Markov's inequality (Proposition 2),

$$\mathbb{E}[X] \geq \frac{\varepsilon^T}{q}\mathbb{P}\left[X > \frac{\varepsilon^T}{q} \,\Big|\, \mathbb{E}[X] \leq \varepsilon^T\right].$$

Since $\varepsilon^T > 0$,

$$q \geq \mathbb{P}\left[X > \frac{\varepsilon^T}{q} \,\Big|\, \mathbb{E}[X] \leq \varepsilon^T\right] \tag{23}$$

$$= 1 - \mathbb{P}\left[X \leq \frac{\varepsilon^T}{q} \,\Big|\, \mathbb{E}[X] \leq \varepsilon^T\right] \tag{24}$$

and thus $\mathbb{P}\left[X \leq \frac{\varepsilon^T}{q} \,\Big|\, \mathbb{E}[X] \leq \varepsilon^T\right] \geq 1 - q$. Finally,

$$\mathbb{P}\left[\mathbb{E}[X] \leq \varepsilon^T, X \leq \frac{\varepsilon^T}{q}\right] = \mathbb{P}\left[X \leq \frac{\varepsilon^T}{q} \,\Big|\, \mathbb{E}[X] \leq \varepsilon^T\right]\mathbb{P}\left[\mathbb{E}[X] \leq \varepsilon^T\right] \tag{25}$$

$$= (1 - q)(1 - p), \tag{26}$$

proving that $\hat{\pi}$ is an $\frac{\varepsilon^T}{q}$ - approximation to a $\mu_{k\text{-of-}N}$ - robust policy with probability $(1-p)(1-q)$. $\qquad\square$

