# OpenReview forum: "Learning to Be Cautious"
_TMLR — Accepted by TMLR_

### Review · Reviewer_r1oU · 2025-06-28

**Summary Of Contributions:**

This work considers how to adapt agents when facing new situations, where the agents need to be cautious. Especially, this work proposes a sequence of tasks in this setting, ranging from contextual bandits to environments that depend on contexts. For all these works, this work shows that the k-of-N counterfactual regret minimization (CFR) with an ensemble of networks can generate policies that adopt cautious behavior.  Extending k-of-N CFR to continuing MDPs, the proposed method is efficient and sound for remaining cautious under uncertainty.

**Audience:**

Yes

**Claims And Evidence:**

Yes

**Requested Changes:**

Please see weaknesses above

**Strengths And Weaknesses:**

Overall, I think the strengths of this work beat the weaknesses.

Strengths:

- The considered setting, learning to Be Cautious when facing novel situations, is indeed important and practical.

- I appreciate the idea to generalize settings ranging from contextual bandits to environments that depend on contexts.

- The theoretical analyses of extending k-of-N CFR for Continuing MDPs in Sec. 5 are clear and solid.

Weaknesses:

- It is better to add a pseudo-code of the proposed method in the paper to help the reader better understand this work.

- Why does Fig.4 mainly consider N=20, and what about other choices of N in these experiments?

- It is required to discuss more works about safe RL [1-2] and contained RL [3-4] that are closely related to this work.

Ref:

[1] Towards Safe Reinforcement Learning via Constraining Conditional Value at Risk

[2] Towards safe reinforcement learning with a safety editor policy

[3] WCSAC: Worst-case soft actor critic for safety-constrained reinforcement learning

[4] Constrained variational policy optimization for safe reinforcement learning

---

> ### Author Response · Authors · 2025-08-05
> **Authors Response to Reviewer r1oU**
>
> We thank the reviewer for their thoughtful comments and constructive feedback, which have helped improve the clarity and depth of the submission.
> - We have moved the **pseudo-code** for our method from the appendix to the main body of the paper, under the Complete Algorithm subsection, to improve accessibility.
> - **Choice of "N" values**: the driving and MNIST experiments were implemented by different co-authors, which led to slight differences in the absolute values of $N$. However, the critical factor is the ratio $\frac{k}{N}$, the degree of caution. This ratio remains consistent across all experiments, using the values $\{1.0, 0.5, 0.1, 0.05\}$. In summary, while the individual values of $k$ and $N$ differ, the caution level remains equivalent, e.g., $\frac{5}{10}$ and $\frac{10}{20}$ yield the same behavior, and our results reflect this consistency.
> - **Safe RL related Work**: [1] uses a CVaR risk measure in a policy optimization. However, their risk is defined exclusively over return in one fixed training environment, and so it only considers risk due to aleatoric uncertainty. The work then explores whether optimizing for risk in aleatoric uncertainty helps with environment perturbations (transition or observation changes), which can be thought of as adding epistemic uncertainty. Their results largely showed little induction of caution under perturbation scenarios due to optimizing for only aleatoric uncertainty.
> Similarly, [2] and [4] operate under constrained MDP (CMDP) frameworks, assuming known constraints and observable rewards during training and evaluation. These methods aim to minimize constraint violations or optimize under constraint satisfaction. Compared to CMDPs, our work aims to make the agent learn to behave cautiously when facing novel situations without relying on explicit constraints. [3] shares with our method the use of CVaR to temper pessimism in worst-case optimization, but assumes that rewards remain defined and learnable during adaptation. Overall, while these works focus on robustness under aleatoric uncertainty or predefined constraints, our approach addresses epistemic-safe reinforcement learning by enabling agents to act cautiously in novel environments without access to reward or constraint supervision. We emphasize zero-shot generalization, where agents must navigate novel tasks with entirely undefined rewards.
> We have added the references [1–4] to our Related Work section, highlighting both the similarities and the key distinctions from our approach.

---

### Review · Reviewer_eKFH · 2025-07-06

**Summary Of Contributions:**

The paper introduces Learning to Be Cautious, a framework that combines an ensemble of neural‐network reward models to quantify epistemic uncertainty and a robust optimization algorithm based on k‑of‑N Counterfactual Regret Minimisation (CFR). By optimising the worst‑k-of-N sampled reward functions, the agent effectively maximises a smooth CVaR‑like risk measure and therefore behaves conservatively in states that are novel with respect to the training data. The authors present three image-based decision-making tasks that progressively remove “obvious” safe actions, a grid-world driving domain that requires sequential planning, and a proof that k-of-N CFR extends from episodic to continuing MDPs. Across all tasks, more risk‑averse settings (small $k/N$) demonstrably yield safer, slower or “help‑seeking” policies without injecting task‑specific safety rules.

**Audience:**

Yes

**Broader Impact Concerns:**

I am aware of no ethical concerns regarding the paper's results. On the other hand, I suggested to the authors in the requested change section that they further improve the broader impact section.

**Claims And Evidence:**

Yes

**Requested Changes:**

- Figure clarity – In Figure 2, the bars overlay individual run markers; consider separating them (e.g., violin plots) so the variance is visually clearer.

- The appendix is a blend of figures and proofs, with one section overlapping with another. The description and discussion of the figures also blurred with the figures themselves. Could you consider separating these two sections more clearly to help the reader locate the main results within each section?

- A brief sensitivity test (e.g., 0.1 vs. 0.5) would indicate whether caution is robust to the choice of +0.25 “help” reward. It also helps if the author can include another sensitivity test on the specific reward setting in the "Ask for help only when it is available" paragraph.

- In the broader‑impact section, the authors might also mention potential misuse that an adversary could hide reward features to trick a cautious agent into excessive conservatism.

**Strengths And Weaknesses:**

**Strengths**

- The problem motivation is clear. Figure 1 and Section 2 vividly illustrate how extrapolative policies collapse when the input distribution shifts. The staged task suite allows the reader to witness caution emerge as the scenarios become progressively more demanding. The overall exposition also shows a clear narrative.

- The optimization algorithm, built on counterfactual regret minimization and robust optimization, rests on a solid theoretical foundation. Section 5 provides regret bounds for continuing MDPs and states the precise assumptions where the authors are concerned with uncertain rewards only, with transitions assumed to be known (no aleatoric uncertainty).

- The ablations on data size show interesting insights. The numerical study on perturbed training data in Section 4 shows that caution surfaces only after the reward ensemble has learned enough to place trust in the “help” action, thereby clarifying how the method’s behaviour hinges on the quality of its epistemic-uncertainty estimates.


**Weaknesses**

- The first point concerns studies on hyperparameter sensitivity. Both k and N are manually selected, as are the ground-truth reward functions in the experiments. Although Figure 2 (p. 6) shows reassuring monotonic trends, practitioners would benefit from an adaptive strategy, such as temporarily reducing $k$ when model uncertainty spikes.

- The comparison is limited to a risk-neutral ensemble average. Omitting stronger safe-RL references—such as distributional RL with CVaR targets, CMDP methods like CPO, or uncertainty-aware MPC—makes it harder to gauge the real significance of the gains. Including at least one modern safety-oriented baseline would ground the empirical claims.

- Combining an ensemble with CVaR is not entirely new, and the main advance here is operationalising the idea via k-of-N CFR and visualising emergent caution. A head-to-head evaluation on quantitative risk metrics (such as expected worst-case loss) would better highlight the paper’s distinct contribution. I note, however, that TMLR’s acceptance criteria does not concern novelty, so this point is largely advisory.

---

> ### Author Response · Authors · 2025-08-05
> **Authors Response to Reviewer eKFH**
>
> We thank the reviewer for their valuable feedback and thoughtful suggestions, which helped improve both the clarity and quality of the paper.
> ### Requested changes:
> - We have updated Figure 2 by adding a **violin plot** to better illustrate the variance across different $k$-of-$N$ runs. In addition to the stars representing individual runs and a horizontal line for the mean.
> - We **organized the appendix**, with separate sections for experiments and proofs, and subsections for each experiment, $k$-of-$N$ convergence analysis, and the sensitivity of the model to the "help" reward.
> - We have conducted a **sensitivity study for the "help" reward**:
>     - In the `learning to ask for help` task, we tested multiple reward values for the "help" arm $\in \{0.1, 0.25, 0.5\}$. Results indicate that when the "help" reward is low (e.g., 0.1), cautious policies tend to select the "help" arm less frequently in both Fashion and E-MNIST environments. Conversely, with a high reward (e.g., 0.5), cautious policies select the "help" arm more consistently, nearly 100% of the time. Importantly, the relative ranking of the cautious policies remains stable across these variations. We now explicitly reference this analysis in the main text and direct readers to Appendix B.1.3 for further figures and tables.
>     - In the `ask for help only when it is available`, we tested the impact of different "help" reward values $\in \{0.02, 0.05, 0.2\}$, when the "help" arm is only available intermittently. The results (now detailed in the appendix) demonstrate that larger rewards increase the frequency of selecting the "help" action when it is available.
> - We have added a sentence to the **broader impact statement** noting a potential risk: an adversary could manipulate uncertainty or obscure reward-relevant features, potentially causing the agent to act with excessive conservatism.
> ### Addressing Weaknesses:
> - **$k$ & $N$ values selection**: In our framework, $k$ and $N$ values are intended to be designer-specified to reflect the desired level of caution, which naturally varies by application (e.g., medical systems vs. video games). While adaptive values based on uncertainty spikes are an interesting direction, they typically assume the availability of reward feedback to evaluate performance and guide adaptation. In contrast, our work focuses on a more challenging setting where reward signals are not available in novel environments.
> - **Safe RL literature**: In the related work section, we discuss CMDP-based methods, which require the designer to predefine safe states, actions, or safety functions to guide behavior. While effective in constrained settings, this requirement can be difficult to scale to open-ended or complex domains, where enumerating all unsafe scenarios is infeasible. We also note that approaches such as distributional RL with CVaR and uncertainty-aware MPC primarily address aleatoric uncertainty—that is, stochasticity inherent in the environment or returns. In contrast, our work focuses on epistemic uncertainty, where the agent lacks knowledge due to novel states or missing rewards. To the best of our knowledge, ours is the first empirical study that explicitly targets robustness to epistemic uncertainty, leading to cautious policies in novel MDPs where the reward is undefined.
> - **Novelty**: While we agree that combining ensembles with CVaR has been explored in prior work, our contribution lies in enabling agents to calibrate their degree of caution in the face of epistemic uncertainty, a setting not directly addressed by traditional CVaR approaches, which typically focus on return distributions under aleatoric uncertainty. Importantly, $k$-of-$N$ CFR approximates the expected worst-case loss, for example, when $k=1$, and $N$ >> $k$ the agent effectively models the worst plausible return across the ensemble.

---

### Review · Reviewer_1hwv · 2025-07-31

**Summary Of Contributions:**

This paper addresses the problem of inducing zero-shot cautious behavior in reinforcement learning agents, focusing specifically on the train-and-deploy setting where agents must act safely in novel, previously unseen environments. The authors propose a method that models epistemic uncertainty over the reward function using deep neural network ensembles and then optimizes policies via k-of-N Counterfactual Regret Minimization (CFR) to be robust to the worst k reward scenarios. This combination leads to policies that learn caution without task-specific safety engineering, as demonstrated across a well-designed progression of tasks - from contextual bandits to a grid-world environment. The paper is further strengthened by a theoretical extension of k-of-N CFR to continuing MDPs, including regret and convergence guarantees. Overall, the method is well-motivated, clearly presented, and supported by insightful experimental results and theoretical analysis. However, the current implementation is limited to tabular settings and assumes transition certainty, which restricts scalability and generality.

**Audience:**

Yes

**Claims And Evidence:**

Yes

**Requested Changes:**

I have outlined my concerns in the weaknesses subsection above. It would further strengthen the paper if authors could include quantitative comparisons with some prior baseline algorithms.

**Strengths And Weaknesses:**

__Strengths__:

- The focus on zero-shot cautious behavior in novel environments is an important aspect for AI safety, especially in train-and-deploy settings.
- The combination of neural network ensembles (to capture uncertainty) and k-of-N CFR (to optimize for risk-sensitive policies) is motivated and well-justified with intuitive reasoning and experiments.
- The progression of tasks is thoughtfully constructed to show that cautious behavior emerges naturally under reward uncertainty. The use of contextual bandit tasks and sequential decision-making environments like gridworld provides strong qualitative evidence.
- The extension of k-of-N CFR to continuing MDPs bridges theoretical robustness guarantees with practical deployment settings.


__Weaknesses__:

- The current implementation of k-of-N CFR is tabular and uses exact policy evaluation, limiting applicability to small or discretized environments. The paper acknowledges this limitation but does not explore approaches to overcome such constraints, particularly in the experimental setup.
- The proposed algorithm assumes known and deterministic transitions, which may not hold in many real-world applications. This limits generalizability.
- This paper does not compare against alternative risk-sensitive or uncertainty-aware baselines (for example, Bayesian RL, POMDPs, distributional RL) in the more complex tasks.
- While the visualizations and action frequency plots demonstrate the emergence of caution, quantitative evaluation of task performance (for example, safety violations, reward tradeoffs) under uncertainty is limited. In addition, training and querying large ensembles approximating the MDP reward functions has a high computational cost.

---

> ### Author Response · Authors · 2025-08-05
> **Authors Response to Reviewer 1hwv**
>
> We thank the reviewer for their valuable feedback and thoughtful suggestions, which aided in strengthening the paper’s structure and coherence.
> ### Addressing Weaknesses:
>
> - **Scalability beyond tabular settings**: As discussed in Section 6, extending our method to function approximation settings is a key future direction. We cited six prior works that successfully apply CFR with function approximation and noted possibilities for approximating worst-case policy evaluation. While this is not yet fully developed in our current submission, we are actively working on integrating such approaches into our framework.
>
> - **Assumption of deterministic and known transitions**: While our method assumes deterministic transitions (Section 3), it does not assume that the transition function is known to the agent. Our implementation is model-free and does not rely on transition dynamics during learning. We acknowledge the limitation of deterministic transitions in Section 6 and note that relaxing this assumption is necessary for broader applicability. We also mention preliminary exploration of uncertainty over transitions and rewards.
>
> - **Lack of risk-sensitive baseline comparisons**: As noted in Section 2, our experiments are conducted in fully observable settings, where the state (e.g., the MNIST image is provided in full), making POMDP baselines inapplicable. Additionally, our method focuses on epistemic uncertainty, not aleatoric uncertainty (which is absent in our deterministic environments). As a result, Bayesian RL and distributional RL methods (typically designed for stochastic environments) would likely behave similarly to our baseline (since there is no aleatoric uncertainty to be robust to). We clarified this in the experimental section of the revised manuscript.
>
> - **Quantitative evaluation & computational power**: Figure 4 (driving grid-world) provides quantitative metrics, including the agent's speed, collision frequency, and collision. However, in contextual bandit settings, explicit safety violations or trade-offs are not defined. Unlike works that assume known safety constraints, our agents do not have access to predefined safety boundaries. Thus, defining meaningful safety metrics in those settings is non-trivial. Nevertheless, our qualitative results (e.g., action frequency plots) clearly demonstrate caution emerging from learned uncertainty. Similarly, there is no trade-off in the reward function to achieve a higher reward while maintaining a low violation rate, like other related work, where they have a pre-defined safety region.
>
> - **Ensemble model computational power**: although ensemble methods are more expensive, their cost does not scale with the size of the state space but rather with the number of reward models $N$ and the number of $k$-of-$N$ CFR iterations (since we require more than $N$ models in total).
> As noted in Section 3, alternative, more computationally efficient techniques like noisy networks or epistemic neural networks could reduce computational overhead for ensembles in future implementations. We added a statement about the ensemble model's computational power.

---

### Decision · Action_Editor_4qQz · 2025-10-08

**Recommendation:** Accept as is

**Additional Comments:**

Comments: The method is clearly described and motivated; the paper shows a staged progression of tasks, from simple contextual settings to an image-based driving gridworld, where more robust configurations yield slower, safer behavior without task-specific safety rules.    The work also extends k-of-N CFR from finite-horizon to continuing MDPs under reward-uncertainty/transition-certainty, providing the needed regret/optimality analysis, which strengthens the theoretical footing. Across reviews, there is broad agreement that the problem is important, the narrative is clear, and the empirical demonstrations align with the stated claims; the new continuing-MDP analysis is a plus. The reviewers have given several comments for improving the paper, and the authors have addressed most of them as part of the review process. Overall, this is a good fit for TMLR.

**Audience:**

Yes

**Audience Explanation:**

I believe that TMLR’s audience, spanning RL theory and practice, will be interested because the paper tackles a common, high-impact deployment scenario: zero-shot behavior in train-and-deploy settings under distribution shift, and demonstrates that “learned caution” can emerge without task-specific safety rules. The staged suite of tasks (from contextual bandits to a sequential driving gridworld) provides a clear empirical through-line, while the k-of-N CFR formulation with reward-uncertainty ensembles is conceptually simple yet practically relevant. This may also motivate others working on safe learning to further develop the approaches proposed in this work.

**Claims And Evidence:**

Yes

**Claims Explanation:**

Based on my reading of the paper, the reviews, and the authors’ response, I believe that the paper’s core claims are backed by a clear empirical approach and a matching theory. The authors clearly define the train-and-deploy, zero-shot caution target and then demonstrate learned caution across progressively harder tasks (contextual bandits -> image-based tasks -> gridworld).  The method used, an ensemble that models epistemic reward uncertainty paired with k-of-N CFR robust optimization, is described clearly and evaluated consistently with that goal. In the grid-world, more robust configurations yield the intended conservative behavior (slower driving and fewer collisions), and ablations show that caution emerges once the reward ensemble has sufficient data, directly supporting the mechanism the paper claims. Finally, the extension of k-of-N CFR to continuing MDPs comes with formal guarantees, reinforcing that the theoretical claims are sound. Overall, the evidence is accurate, convincing, and clearly presented; the paper can be strengthened in multiple ways, and those concrete suggestions are listed in the additional comments.